# ERROR-FEEDBACK MEETS STOCHASTIC APPROXIMATION WITH TWO TIME SCALES

## ABSTRACT

Two-time-scale stochastic approximation is a recursive algorithm for solving a system of two equations. The method has found broad applications in many areas including machine learning and reinforcement learning. Recent works have revealed that single-time-scale stochastic approximation (especially its variant stochastic gradient descent in optimization) is robust to structured perturbations such as compression, local updates, and delays, but it is not well-understood in the two-time-scale case. Almost nothing is known about the analogous question: Is two-time-scale stochastic approximation also robust to similar structured perturbations? In this paper, we study error-feedback-based two-time-scale stochastic approximation. We propose a unified theory of two-time-scale stochastic approximation based on error-feedback to analyze the impact of different forms of structured perturbations. We show that two-time-scale stochastic approximation is robust to structured perturbations. In particular, two-time-scale stochastic approximation with different forms of structured perturbations exhibits the same non-asymptotic theoretical guarantees as its single-time-scale counterpart without structured perturbations. We further show that the convergence rate in all cases consists of two terms, where only the higher-order term is affected by structured perturbations. This is especially important for distributed parallel implementations of two-time-scale stochastic approximation algorithms.

## 1 INTRODUCTION

Stochastic approximation (SA) is a general class of recursive methods for finding roots of unknown functions for which only noisy accesses are available Robbins & Monro (1951); Borkar (2009). Specifically, the SA method seeks to find $x^*$ such that $h(x^*) = 0$ with the following update:

$$x_{k+1} = x_k + \alpha_k \left( h(x_k) + \xi_k \right), \tag{1}$$

where $\alpha_k$ is the step size and $\xi_k$ is a random variable. Such method has found broad applications in many areas such as machine learning (ML), statistics, stochastic control, and signal processing. In particular, stochastic gradient descent (SGD), a variant of SA, lies at the core of machine learning, especially deep learning Bottou et al. (2018). Notably, the practical success of SA, especially SGD, can be attributed to its robustness to robust to structured perturbations such as compression, local updates, and delays Stich & Karimireddy (2020). This is especially important for distributed parallel implementations in the sense that a parallel version of SGD with structured perturbations can efficiently use the computing power of multiple parallel agents. For instance, local SGD Stich (2018), a variant of SGD, allows update (1) to evolve locally on each agent, independently of each other, and only average the sequence once in a while. The results show that local SGD is as computationally efficient as parallel mini-batch SGD, but the communication cost can be significantly reduced.

While most SA studies have focused on the single sequence case, two-time-scale SA (TTSA) was introduced in Robbins & Monro (1951), and it has been widely applied to problems involving two coupled sequence updates. Specifically, given two nonlinear operators $f : \mathbb{R}^{d_0} \times \mathbb{R}^{d_1} \to \mathbb{R}^{d_0}$ and $g : \mathbb{R}^{d_0} \times \mathbb{R}^{d_1} \to \mathbb{R}^{d_1}$, the TTSA method aims to solve a system of two equations:

$$\begin{cases} f(x, y) = 0, \\ g(x, y) = 0, \end{cases} \tag{2}$$

by two coupled sequence updates of the form:

$$x_{k+1} = x_k + \alpha_k \left( f(x_k, y_k) + \xi_k \right), \tag{3}$$

$$y_{k+1} = y_k + \beta_k \left( g(x_k, y_k) + \psi_k \right), \tag{4}$$

where $\alpha_k, \beta_k$ are step sizes and $\xi_k, \psi_k$ are random variables. The TTSA method has found broad applications in many areas including machine learning and reinforcement learning. In particular, the TTSA method has been studied mostly in the context of stochastic bilevel optimization (SBO) Ruszczynski (2021); Balasubramanian et al. (2022) and stochastic compositional optimization (SCO) Wang et al. (2017); Gao & Huang (2021); Jiang et al. (2022) where many typical SBO and SCO algorithms are exactly in the form of (3)-(4). It is worthwhile mentioning that the SBO and SCO problems encompass many contemporary ML problems including adversarial robustness, hyperparameter tuning, meta-learning, reinforcement learning; see e.g., Franceschi et al. (2018); Zhang et al. (2022); Hong et al. (2023). While recent work has begun to study SBO and SCO in distributed parallel settings Tarzanagh et al. (2022); Yang et al. (2022), a generic theory for distributed TTSA is less developed. Almost nothing is known about the question:

> *Is two-time-scale stochastic approximation robust to structured perturbations such as compression, local updates, and delays?*

To fill this gap, we study error-feedback-based TTSA. Error-feedback is a unified framework for analyzing the theory of SGD with different forms of structured perturbation Stich et al. (2018); Karimireddy et al. (2019); Stich & Karimireddy (2020). In this work, we extend the error-feedback framework to TTSA. Indeed, analyzing error-feedback-based TTSA is challenging as it involves two coupled sequences for a one-step update in TTSA, and there are complicated interactions between error-feedback and TTSA. Two auxiliary sequences are introduced in the error-feedback framework, which aggregate the structured perturbation errors. Moreover, existing works on error-feedback only consider variants of SGD and do not consider SA, let alone TTSA. Notice that SA is a more general class of algorithms that covers many algorithms in reinforcement learning that cannot be formulated as SGD and its variants Kaledin et al. (2020); Chen et al. (2022).

### 1.1 MAIN CONTRIBUTIONS

Our main contributions are summarized as follows:

**1) Error-feedback meets two-time-scale stochastic approximation.** We give an affirmative answer to the above question and present a framework for error-feedback-based two-time-scale stochastic approximation (EF-TTSA) that captures a rich class of structured perturbations such as compression, local updates, and delays. We utilize the framework to analyze the effect of different forms of structured perturbations on EF-TTSA in a unified manner. To the best of our knowledge, this is the first work that considers two-time-scale stochastic approximation corrupted by structured perturbations with theoretical convergence guarantees.

**2) Error-compensated TTSA with arbitrary compressors.** We propose an instance of EF-TTSA, error-compensated TTSA with arbitrary compressors (Algorithm 1), in which compression operators are used to reduce communication costs. We prove that our Algorithm 1 attains an $\mathcal{O}\left(\frac{1}{T} + \frac{1}{\delta^2 T^2}\right)$ convergence rate, where $T$ is the total number of iterations, and $\delta$ is the compressed parameter. We see that the compression operator only affects the higher-order term of the convergence rate. Thus, the effects of the compression become negligible after a few iterations and the algorithm converges at the same rate as standard TTSA without compression Shen & Chen (2022).

**3) Local TTSA with periodic global averaging.** We propose an instance of EF-TTSA, local TTSA with periodic global averaging (Algorithm 2), in which agents perform multiple local iterative updates, followed by global averaging. We prove that our Algorithm 2 attains an $\mathcal{O}\left(\frac{1}{T} + \frac{K^2}{T^2}\right)$ convergence rate, where $K$ is the communication interval. We also observe that the effects of multiple local updates become negligible after a few iterations, suggesting that our algorithm gains communication efficiency through infrequent communication, essentially for free.

**4) TTSA with delayed updates.** We propose an instance of EF-TTSA, TTSA with delayed updates (Algorithm 3), where updates are delayed and reflect iterations from $\tau$ rounds ago. We prove that our Algorithm 3 attains an $\mathcal{O}\left(\frac{1}{T} + \frac{\tau^2}{T^2}\right)$ convergence rate. Similarly, $\tau$ only appears in the higher-order

term of the convergence rate, and its effect becomes negligible when $T$ is large enough. The results show that the performance of TTSA with delays is comparable to that of TTSA without delays.

## 1.2 RELATED WORK

### 1.2.1 TWO-TIME-SCALE STOCHASTIC APPROXIMATION

TTSA, a generalized variant of SA, has been studied for a long time. Specifically, the asymptotic behavior of TTSA has been analyzed in Borkar (1997) using an ODE approach, and in Tadic & Meyn (2003) under Markovian noise. Recent work has heavily focused on the finite-time performance of TTSA for both linear Konda & Tsitsiklis (2004); Doan & Romberg (2019); Kaledin et al. (2020); Doan (2021b) (when $f$ and $g$ are linear functions with respect to their variables) and nonlinear settings Dalal et al. (2018); Zeng et al. (2021); Doan (2022), under both i.i.d. and Markovian samples. All of these works use the so-called fast and slow time scales: one sequence is updated in the fast-time scale while the other is in the slow time scale; the time scale difference $\lim_{k\to\infty} \alpha_k/\beta_k \to 0$. With the proper choice of step sizes, the two sequences with the fast and slow time scales are asymptotically decoupled. Shen & Chen (2022) established an improved analysis of nonlinear TTSA in which the two sequences are updated in the same time scale, i.e., $\lim_{k\to\infty} \alpha_k/\beta_k = c$ for some constant $c > 0$. Shen & Chen (2022) demonstrated that the sequences generated by nonlinear TTSA converge to desired solutions at a tight rate $\mathcal{O}(\frac{1}{T})$ for the strongly-monotone case. Distributed variants of linear and nonlinear TTSA are considered in Doan & Romberg (2020) and Doan (2021a), respectively. Doan (2021a) studied the convergence rate of distributed local TTSA; however, the sequences generated by the method only converge linearly to a ball encircled the desired solution. In summary, it is an open question whether convergence guarantees for TTSA with structured perturbations (e.g., compression, local updates, and delays) can be achieved.

### 1.2.2 ERROR-FEEDBACK FRAMEWORK

Error-feedback relates closely to communication-efficient methods such as quantization and sparsification in distributed optimization literature. Roughly speaking, error-feedback is a memory mechanism that uses accumulated errors from previous iterations for bias correction. The idea of error-feedback was introduced in Seide et al. (2014) to study 1-bit SGD, aiming to counter the effect of bias introduced by quantization. Since then, several papers Alistarh et al. (2018); Stich et al. (2018); Karimireddy et al. (2019); Lin et al. (2022) considered compression methods with error-feedback, i.e., incorporating the error made by the compression operator to correct the current direction. For instance, Stich et al. (2018); Karimireddy et al. (2019) demonstrated that SignSGD with error-feedback, a very aggressive compression method where each coordinate of the gradient is replaced by its sign, retains almost the same behavior as SGD without compression.

For further reducing communication costs, various orthogonal techniques have been proposed, such as asynchrony (delayed updates) Stich (2018) and periodic averaging (local updates) Arjevani et al. (2020) in distributed optimization literature. Stich & Karimireddy (2020) presented a framework for sgd with error-feedback, and analyzed the effect of different forms of structured perturbations. In particular, SGD with delayed updates and SGD with local updates essentially act like compressed SGD with error-feedback. Mitra et al. (2023) analyzed compressed temporal difference learning with error-feedback, and proved that temporal difference learning is robust to structured perturbations; but the author only studied compression. In this work, we use the error-feedback framework to analyze nonlinear TTSA with different structured perturbations in a unified manner.

## 2 PRELIMINARIES

We are interested in the two-time-scale SA problem (2) under the following assumptions.

**Assumption 1.** *For any $x \in \mathbb{R}^{d_0}$, there exists a unique $y^*(x) \in \mathbb{R}^{d_1}$ such that $g(x, y^*(x)) = 0$. Moreover, there exist $L_{y,0}$ and $L_{y,1}$ such that for any $x_1, x_2 \in \mathbb{R}^{d_0}$, the following inequalities hold*

$$||y^*(x_1) - y^*(x_2)|| \le L_{y,0}||x_1 - x_2||, \tag{5}$$

$$||\nabla y^*(x_1) - \nabla y^*(x_2)|| \le L_{y,1}||x_1 - x_2||. \tag{6}$$

**Assumption 2.** *For any $x_1$, $x_2 \in \mathbb{R}^{d_0}$, and $y_1$, $y_2 \in \mathbb{R}^{d_1}$, there exist $L$, $L_f$ and $L_g$ such that*

$$||f(x_1, y^*(x_1)) - f(x_2, y^*(x_2))|| \leq L||x_1 - x_2||, \tag{7}$$

$$||f(x_1, y_1) - f(x_2, y_2)|| \leq L_f \left(||x_1 - x_2|| + ||y_1 - y_2||\right), \tag{8}$$

$$||g(x_1, y_1) - g(x_2, y_2)|| \leq L_g \left(||x_1 - x_2|| + ||y_1 - y_2||\right). \tag{9}$$

**Assumption 3.** *Suppose $f(x, y)$ is one-point strongly monotone on $x^*$; that is, there exists a constant $\lambda_f > 0$ such that*

$$\langle x - x^*, f(x, y^*(x)) \rangle \leq -\lambda_f ||x - x^*||^2. \tag{10}$$

*Moreover, suppose $g(x, y)$ is one-point strongly monotone on $y^*(x)$ for any given $x \in \mathbb{R}^{d_0}$; that is, there exists a constant $\lambda_g > 0$ such that*

$$\langle y - y^*(x), g(x, y) \rangle \leq -\lambda_g ||y - y^*(x)||^2. \tag{11}$$

**Remark 1.** *Assumptions 1-3 are fairly standard in the analysis of two-time-scale SA; see e.g., Mokkadem & Pelletier (2006); Kaledin et al. (2020); Zeng et al. (2021); Shen & Chen (2022).*

### 2.1 MOTIVATING APPLICATION EXAMPLES

#### 2.1.1 STOCHASTIC BILEVEL OPTIMIZATION

With mappings $F : \mathbb{R}^{d_0} \times \mathbb{R}^{d_1} \to \mathbb{R}$ and $G : \mathbb{R}^{d_0} \times \mathbb{R}^{d_1} \to \mathbb{R}$, the stochastic bilevel optimization (SBO) problem can be formulated as

$$\min_{x \in \mathbb{R}^{d_0}} F(x, y) := \mathbb{E}_\theta[F(x, y(x); \theta)], \text{ s.t. } y(x) := \arg \min_{y \in \mathbb{R}^{d_1}} G(x, y) := \mathbb{E}_\zeta[G(x, y; \zeta)]. \tag{12}$$

A class of gradient-based methods is a popular approach to solve problem (12); see e.g., Ghadimi & Wang (2018); Chen et al. (2021b). In particular, this type of method has updates

$$x_{k+1} = x_k + \alpha_k \left(\nabla_x F(x_k, y_k; \theta_k) - \nabla_{xy}^2 G(x_k, y_k; \zeta_k) H_{yy}(x_k, y_k; \zeta_k') \nabla_y F(x_k, y_k; \theta_k)\right), \tag{13}$$

$$y_{k+1} = y_k - \beta_k \nabla_y G(x_k, y_k; \zeta_k''), \tag{14}$$

where $H_{yy}(x_k, y_k; \zeta_k')$ is a stochastic approximation of the Hessian inverse $[\nabla_{yy} G(x_k, y_k)]^{-1}$. We observe that the update (13)-(14) is a special case of the TTSA update in (3)-(4) by defining:

$$f(x_k, y_k) = \nabla_x F(x_k, y_k) - \nabla_{xy}^2 G(x_k, y_k)[\nabla_{yy} G(x_k, y_k)]^{-1} \nabla_y F(x_k, y_k), \tag{15}$$

$$\xi_k = -f(x_k, y_k) + \nabla_x F(x_k, y_k; \theta_k) - \nabla_{xy}^2 G(x_k, y_k; \zeta_k) H_{yy}(x_k, y_k; \zeta_k') \nabla_y F(x_k, y_k; \theta_k), \tag{16}$$

$$g(x_k, y_k) = -\nabla_y G(x_k, y_k), \quad \psi_k = -g(x_k, y_k) - \nabla_y G(x_k, y_k; \zeta_k''), \tag{17}$$

Shen & Chen (2022) demonstrated that the standard conditions in the SBO literature Ghadimi & Wang (2018); Chen et al. (2021b) lead to Assumptions 1-3 in this work.

#### 2.1.2 STOCHASTIC COMPOSITIONAL OPTIMIZATION

With outer function $F(y; \theta) : \mathbb{R}^{d_1} \to \mathbb{R}$ and inner function $G(x; \zeta) : \mathbb{R}^{d_2} \to \mathbb{R}^{d_1}$, the stochastic compositional optimization (SCO) problem can be formulated as

$$\min_{x \in \mathbb{R}^{d_2}} F(G(x)) := \mathbb{E}_\theta[F(G(x); \theta)], \text{ with } G(x) := \mathbb{E}_\zeta[G(x; \zeta)]. \tag{18}$$

To solve (18), a popular method Yang et al. (2019) takes the following form

$$x_{k+1} = x_k - \alpha_k \nabla F(y_k; \theta_k) \nabla G(x_k; \zeta_k), \tag{19}$$

$$y_{k+1} = y_k + \beta_k \left(F(y_k; \theta_k') - y_k\right), \tag{20}$$

where $y_k$ is used to directly track $\mathbb{E}_\zeta[G(x; \zeta)]$. We observe that the update (19)-(20) is a special case of the TTSA update in (3)-(4) by defining:

$$f(x_k, y_k) = -\nabla F(y_k) \nabla G(x_k), \quad \xi_k = -f(x_k, y_k) - \nabla(y_k, \theta_k) \nabla G(x_k; \zeta_k), \tag{21}$$

$$g(x_k, y_k) = F(y_k) - y_k, \quad \psi_k = -g(x_k, y_k) + F(y_k; \theta_k') - y_k. \tag{22}$$

Likewise, Shen & Chen (2022) demonstrated that the standard conditions in the SCO literature Yang et al. (2019); Chen et al. (2021a) ensure that Assumptions 1-3 in this work hold.

## 3 ERROR-FEEDBACK MEETS TWO-TIME-SCALE SA

In this section, we introduce our framework EF-TTSA wherein two sequences $\{x_k\}$, $\{y_k\}$ and two auxiliary sequences $\{d_k\}$, $\{e_k\}$ all evolve at the same time, using the following expressions:

$$x_{k+1} = x_k + \mu_k, \tag{23}$$
$$d_{k+1} = d_k + \alpha_k f_k - \mu_k, \tag{24}$$
$$y_{k+1} = y_k + \nu_k, \tag{25}$$
$$e_{k+1} = e_k + \beta_k g_k - \nu_k, \tag{26}$$

with $d_0 = e_0 = 0$. In (23)-(26), $\{\mu_k\}_{k\geq 0}$ and $\{\nu_k\}_{k\geq 0}$ are two sequences, representing the updates applied to $\{x_k\}_{k\geq 0}$ and $\{y_k\}_{k\geq 0}$ respectively. $\{d_k\}_{k\geq 0}$ and $\{e_k\}_{k\geq 0}$ aggregate the structured perturbation errors. We denote by $f_k = f(x_k, y_k) + \xi_k$ and $g_k = g(x_k, y_k) + \psi_k$, where $\xi_k$ and $\psi_k$ are two independent random variables. Note that our framework EF-TTSA is generic and covers many special cases: error-compensated TTSA with arbitrary compressors (cf. Section 4), local TTSA with periodic global averaging (cf. Section 5), and TTSA with delayed updates (cf. Section 6).

We define the filtration $\mathcal{F}_k = \{x_0, y_0, x_1, y_1, \cdots, x_k, y_k\}$. Regarding the random variables $\{\xi_k, \psi_k\}_{k\geq 0}$, we impose the following standard assumption Doan (2022).

**Assumption 4.** *The random variables $\xi_\psi$, $\psi_\xi$, for all $k \geq 0$, are independent of each other and across time. Moreover, there exist two positive constants $\sigma_\xi$, $\sigma_\psi$ such that*

$$\mathbb{E}\left[\xi_k | \mathcal{F}_k\right] = 0, \ \mathbb{E}\left[\psi_k | \mathcal{F}_k\right] = 0, \ ||\xi_k|| \leq \sigma_\xi, \ ||\psi_k|| \leq \sigma_\psi, \quad \forall k \geq 0. \tag{27}$$

For the convenience of analysis, define $\bar{x}_k = x_k + d_k$, $\bar{y}_k = y_k + e_k$, $y_k^* = y^*(x_k)$, $\bar{y}_k^* = y^*(\bar{x}_k)$, and $\Xi_k = \mathbb{E}[||\bar{y}_k - \bar{y}_k^*||^2 + ||\bar{x}_k - x^*||^2]$. By the definitions of $\bar{x}_k$ and $\bar{y}_k$, we have that $\bar{x}_{k+1} = \bar{x}_k + \alpha_k f_k$; $\bar{y}_{k+1} = \bar{y}_k + \beta_k g_k$. The main result of this section is as follows.

**Lemma 1.** *Let $\{x_k, d_k, y_k, e_k\}_{k\geq 0}$ be the sequence generated by (23)-(26). Suppose Assumptions 1-4 hold, and set $\alpha_k \leq \min\{\frac{\lambda_f}{16L_y L^2}, \frac{\lambda_g^2}{6L_g^2(2c_1+\lambda_f)}\}$ and $\beta_k = \bar{\beta}\alpha_k$ where $\bar{\beta} = \frac{2c_1+\lambda_f}{\lambda_g}$, we have*

$$\Xi_{k+1} \leq (1 - \frac{\lambda_f}{2}\alpha_k)\Xi_k + \Delta_1\alpha_k(||d_k||^2 + ||e_k||^2) + \Delta_2\alpha_k^2(\sigma_\xi^2 + \sigma_\psi^2), \tag{28}$$

*where $\Delta_1 = (1 + c_2)\left(3L_g^2 + 2L_g^2/\lambda_g\right)\bar{\beta} + c_3$, $\Delta_2 = (1 + c_2)\bar{\beta}^2 + L_y$, $c_1 = L_{y,0}L + 4L_{y,0}L_f + 2L_{y,1}\sigma_\xi^2 + 4L_y L_f^2 + \frac{4L_{y,0}^2 L^2 + 9L_f^2}{\lambda_f}$, $c_2 = L_{y,0}L_f + L_{y,0}L + 2L_{y,1}\sigma_\xi^2 + \frac{4L_{y,0}^2 L^2}{\lambda_f}$, $c_3 = L_{y,0}L + 3L_{y,0}L_y L_f + 4L_y L_f^2 + \frac{12L_f^2 L_y}{\lambda_f}$, and $L_y = L_{y,0}^2 + L_{y,1} + 1$.*

Note that the above lemma derives an upper bound on the one-step progress of $\Xi_k$, which is key to our analysis and will simplify the presentation of the proofs in the subsequent sections. Observe that there is an "error" term on the right side of the inequality (28): $||d_k||^2 + ||e_k||^2$ measures the mismatch between the true sequences $\{x_k\}$, $\{y_k\}$ and their noisy estimates $\{\bar{x}_k\}$, $\{\bar{y}_k\}$.

## 4 ERROR-COMPENSATED TTSA WITH ARBITRARY COMPRESSORS

In this section, we propose an instance of EF-TTSA, error-compensated TTSA with arbitrary compressors. First, we use the EF-TTSA framework to analyze error-compensated TTSA with arbitrary compressors, and then extend the results to the SBO and SCO problems.

---

**Algorithm 1** Error-Compensated TTSA with Arbitrary Compressors

---

1: **Initialization:** $\{\alpha_k\}_{k\geq 0}, \{\beta_k\}_{k\geq 0}, x_0 \in \mathbb{R}^{d_0}, y_0 \in \mathbb{R}^{d_1}, d_{0,i} = e_{0,i} = 0, \forall i \in [n]$
2: **for** $k = 0, 1, \cdots, T-1$ **do**
3:    **for each agent** $i \in [n]$ **do**
4:       $\mu_{k,i} = \mathcal{Q}[d_{k,i} + \alpha_k f_{k,i}]$
5:       $d_{k+1,i} = d_{k,i} + \alpha_k f_{k,i} - \mu_{k,i}$
6:       $\nu_{k,i} = \mathcal{Q}[e_{k,i} + \beta_k g_{k,i}]$
7:       $e_{k+1,i} = e_{k,i} + \beta_k g_{k,i} - \nu_{k,i}$
8:    **end for**
9:    **on server**
10:       $x_{k+1} = x_k + \frac{1}{n}\sum_{i\in[n]}\mu_{k,i}, \quad y_{k+1} = y_k + \frac{1}{n}\sum_{i\in[n]}\nu_{k,i}$
11: **end for**

---

The error-compensated TTSA with arbitrary compressors is illustrated in Algorithm 1. Each agent $i \in [n]$ stores and updates local sequence $\{\mu_{k,i}, d_{k,i}, \nu_{k,i}, e_{k,i}\}$ and communicates with the central server to update global sequence $\{x_k, y_k\}$. Following the MEM-SGD Stich et al. (2018), the sequence $\{\mu_{k,i}, d_{k,i}, \nu_{k,i}, e_{k,i}\}$ is updated in the following way:

$$\mu_{k,i} = \mathcal{Q}[d_{k,i} + \alpha_k f_{k,i}], \quad d_{k+1,i} = d_{k,i} + \alpha_k f_{k,i} - \mu_{k,i}, \tag{29}$$

$$\nu_{k,i} = \mathcal{Q}[e_{k,i} + \beta_k g_{k,i}], \quad e_{k+1,i} = e_{k,i} + \beta_k g_{k,i} - \nu_{k,i}, \tag{30}$$

where for any agent $i \in [n]$ and $k \geq 0$, $f_{k,i} = f(x_k, y_k) + \xi_{k,i}, g_{k,i} = g(x_k, y_k) + \psi_{k,i}$, and $\mathcal{Q}[\cdot]$ is a compression operator that satisfies the following contraction property.

**Assumption 5.** *The compression operator $\mathcal{Q} : \mathbb{R}^d \to \mathbb{R}^d$ satisfies the following inequality*

$$\mathbb{E}_{\mathcal{Q}}[||\mathcal{Q}[x] - x||^2] \leq (1-\delta)||x||^2, \tag{31}$$

*for a parameter $\delta \geq 0$ and $\forall x \in \mathbb{R}^d$. Here $\mathbb{E}_{\mathcal{Q}}[\cdot]$ denotes the expectation over the randomness of $\mathcal{Q}$.*

As for the central server, when it receives $\{\mu_{k,i}\}$ and $\{\nu_{k,i}\}$ from all agents $[n]$, it updates the global sequence $\{x_k, y_k\}$ as $x_{k+1} = x_k + \frac{1}{n}\sum_{i\in[n]}\mu_{k,i}, y_{k+1} = y_k + \frac{1}{n}\sum_{i\in[n]}\nu_{k,i}$, and then sends $x_{k+1}$ and $y_{k+1}$ back to all agents, as shown in line 10 of Algorithm 1.

Note that instead of transmitting full-dimensional vectors, Algorithm 1 improves communication efficiency by using limited bit representation (quantization) or enforcing sparsity. Observe that Algorithm 1 takes the following form in the EF-TTSA framework:

$$d_k = \frac{1}{n}\sum_{i\in[n]} d_{k,i}, \quad \mu_k = \frac{1}{n}\sum_{i\in[n]} \mu_{k,i}, \quad e_k = \frac{1}{n}\sum_{i\in[n]} e_{k,i}, \quad \nu_k = \frac{1}{n}\sum_{i\in[n]} \nu_{k,i}. \tag{32}$$

In view of (32), applying Lemma 1, we have

$$\Xi_{k+1} \leq (1 - \frac{\lambda_f}{2}\alpha_k)\Xi_k + \Delta_1\alpha_k\Phi_k + \Delta_2\alpha_k^2(\sigma_\xi^2 + \sigma_\psi^2), \tag{33}$$

where $\Xi_k = \mathbb{E}[||\bar{y}_k - y^*(\bar{x}_k)||^2 + ||\bar{x}_k - x^*||^2], \Phi_k = \mathbb{E}\left[\frac{1}{n}\sum_{i\in[n]}||d_{k,i}||^2 + \frac{1}{n}\sum_{i\in[n]}||e_{k,i}||^2\right]$, $\bar{x}_k = x_k + \frac{1}{n}\sum_{i\in[n]} d_{k,i}$, and $\bar{y}_k = y_k + \frac{1}{n}\sum_{i\in[n]} e_{k,i}$. We then derive an upper bound on $\Phi_k$.

**Lemma 2.** *Let $\{x_k, d_{k,i}, y_k, e_{k,i}\}$ be the sequence generated by Algorithm 1. Suppose Assumptions 1-5 hold and set $\alpha_k \leq \frac{\delta}{\sqrt{20(4L_f^2 + 3L_g^2\bar{\beta}^2)}}$ and $\beta_k = \bar{\beta}\alpha_k$, we have*

$$\Phi_{k+1} \leq (1 - \frac{\delta}{2})\Phi_k + (1 + \frac{4}{\delta})(4L_f^2 + 4L^2 + 3L_g^2\bar{\beta}^2)\alpha_k^2\Xi_k + (1 + \bar{\beta}^2)\alpha_k^2(\sigma_\xi^2 + \sigma_\psi^2). \tag{34}$$

We now obtain the main theorem for error-compensated TTSA with arbitrary compressors.

**Theorem 1.** *Consider the sequence $\{x_k, y_k\}$ generated by Algorithm 1. Suppose Assumptions 1-5 hold. Selecting step sizes $\alpha_k = \Theta(\frac{1}{k+1/\delta})$ and $\beta_k = \Theta(\frac{1}{k+1/\delta})$, then it holds that*

$$\frac{1}{W_T}\sum_{k=0}^{T-1} w_k\mathbb{E}[||y_k - y^*(x_k)||^2 + ||x_k - x^*||^2] \leq \mathcal{O}\left(\frac{1}{T} + \frac{1}{\delta^2 T^2}\right), \tag{35}$$

*for some sequence of positive weights $w_k = k + \kappa$ where $\kappa \geq \frac{16}{\delta}$ and $W_T = \sum_{k=0}^{T} w_k$. As a consequence, it holds that $\lim_{k \to \infty} ||y_k - y^*(x_k)||^2 = 0$ a.s. and $\lim_{k \to \infty} ||x_k - x^*||^2 = 0$ a.s.*

**Remark 2.** *Theorem 1 implies that the compression operator only affects the higher order term of the convergence rate. When $T$ is sufficiently large, the first term is dominating in (35), and the error-compensated TTSA with arbitrary compressors converges at a tight rate $\mathcal{O}(1/T)$, which recovers the convergence rate of nonlinear TTSA with exact communication.*

**Remark 3.** *We can extend Theorem 1 to the SBO and SCO problems. Consider the SBO algorithm with the updates in (13)-(14) and the SCO algorithm with the updates in (19)-(20). Assuming Assumptions 1-5 holds and selecting step sizes $\alpha_k = \Theta(\frac{1}{k+1/\delta})$ and $\beta_k = \Theta(\frac{1}{k+1/\delta})$, the resulting error-compensated SBO and SCO with arbitrary compressors can converge at rate $\mathcal{O}\left(\frac{1}{T} + \frac{1}{\delta^2 T^2}\right)$.*

## 5 LOCAL TTSA WITH PERIODIC GLOBAL AVERAGING

In this section, we propose the second instance of EF-TTSA, namely, local TTSA with periodic global averaging. First, we use the EF-TTSA framework to analyze local TTSA with periodic global averaging, and then extend the results to the SBO and SCO problems.

---

**Algorithm 2** Local TTSA with Periodic Global Averaging

---

1: **Initialization:** $\{\alpha_k\}_{k \geq 0}, \{\beta_k\}_{k \geq 0}, x_0 \in \mathbb{R}^{d_0}, y_0 \in \mathbb{R}^{d_1}$
2: **for** $k = 0, 1, \cdots, T - 1$ **do**
3:    **for each agent** $i \in [n]$ **do**
4:       $x_{k+1,i} = x_{k,i} + \alpha_k f_{k,i}$
5:       $y_{k+1,i} = y_{k,i} + \beta_k g_{k,i}$
6:    **end for**
7:    **if** $\mod(k + 1, K) = 0$ **then**
8:       $x_{k+1,i} = \frac{1}{n} \sum_{j \in [n]} x_{k+1,j}, \quad y_{k+1,i} = \frac{1}{n} \sum_{j \in [n]} y_{k+1,j}$
9:    **end if**
10: **end for**

---

The local TTSA with periodic global averaging is illustrated in Algorithm 2. Following the local SGD Stich (2018), the algorithm evolves agents $[n]$ and sequences $\{x_{k,i}, y_{k,i}\}_{i \in [n]}$ in parallel. Specifically, the sequence $\{x_{k,i}, y_{k,i}\}$ is updated in the following way:

$$x_{k+1,i} = x_{k,i} + \alpha_k f_{k,i}, \tag{36}$$
$$y_{k+1,i} = y_{k,i} + \beta_k g_{k,i}, \tag{37}$$

where for any agent $i \in [n]$ and $k \geq 0$, $f_{k,i} = f(x_{k,i}, y_{k,i}) + \xi_{k,i}$, and $g_{k,i} = g(x_{k,i}, y_{k,i}) + \psi_{k,i}$. If $\mod(k + 1, K) = 0$, the central server is responsible to synchronize the sequences:

$$x_{k+1,i} = \frac{1}{n} \sum_{j \in [n]} x_{k+1,j}, \quad y_{k+1,i} = \frac{1}{n} \sum_{j \in [n]} y_{k+1,j}, \quad \forall i \in [n]. \tag{38}$$

Note that instead of communicating at every iteration, Algorithm 2 achieves communication reduction by allowing multiple local updates (i.e., reducing communication frequency). Observe that Algorithm 2 takes the following form in the EF-TTSA framework:

$$d_{k,i} = \bar{x}_k - x_{k,i}, \quad \mu_{k,i} = \begin{cases} \alpha_k f_{k,i} & \text{if } \mod(k+1, K) \neq 0, \\ \bar{x}_k - x_{k,i} + \alpha_k f_k & \text{otherwise,} \end{cases} \tag{39}$$

$$e_{k,i} = \bar{y}_k - y_{k,i}, \quad \nu_{k,i} = \begin{cases} \beta_k g_{k,i} & \text{if } \mod(k+1, K) \neq 0, \\ \bar{y}_k - y_{k,i} + \beta_k g_k & \text{otherwise,} \end{cases} \tag{40}$$

where $\bar{x}_k = \frac{1}{n} \sum_{i \in [n]} x_{k,i}, \bar{y}_k = \frac{1}{n} \sum_{i \in [n]} y_{k,i}, f_k = \frac{1}{n} \sum_{i \in [n]} f_{k,i}$, and $g_k = \frac{1}{n} \sum_{i \in [n]} g_{k,i}$.

In view of (39)-(40), applying Lemma 1, we have

$$\Xi_{k+1} \leq (1 - \frac{\lambda_f}{2} \alpha_k) \Xi_k + \Delta_1 \alpha_k \Phi_k + \Delta_2 \alpha_k^2 (\sigma_\xi^2 + \sigma_\psi^2), \tag{41}$$

where $\Phi_k = \mathbb{E}\left[\frac{1}{n}\sum_{i\in[n]}||\bar{x}_k - x_{k,i}||^2 + \frac{1}{n}\sum_{i\in[n]}||\bar{y}_k - y_{k,i}||^2\right]$ measures the deviation of the local sequences $\{x_{k,i}\}$ and $\{y_{k,i}\}$. The next result establishes an upper bound on the deviation $\Phi_k$.

**Lemma 3.** *Let $\{x_{k,i}, d_{k,i}, y_{k,i}, e_{k,i}\}$ be the sequence generated by Algorithm 2, and $k_0 = \lfloor k/K \rfloor$. Suppose Assumptions 1-4 hold, and set $\alpha_k \leq \frac{1}{\sqrt{(4L_f^2 + 4L^2 + 3L_g^2\bar{\beta}^2)K}}$ and $\beta_k = \bar{\beta}\alpha_k$, we have*

$$\Phi_k \leq 2(4L_f^2 + 3L_g^2\bar{\beta}^2)K\sum_{j=k_0}^{k-1}\alpha_j^2\Xi_j + 2(1+\bar{\beta}^2)\sum_{j=k_0}^{k-1}\alpha_j^2\left(\sigma_\xi^2 + \sigma_\psi^2\right). \tag{42}$$

We are now ready to present the main theorem for local TTSA with periodic global averaging.

**Theorem 2.** *Consider the sequence $\{x_{k,i}, y_{k,i}\}$ generated by Algorithm 2. Suppose Assumptions 1-4 hold. Selecting step sizes $\alpha_k = \Theta(\frac{1}{k+K})$ and $\beta_k = \Theta(\frac{1}{k+K})$, then it holds that*

$$\frac{1}{W_T}\sum_{k=0}^{T-1}w_k\mathbb{E}[||\bar{y}_k - y^*(\bar{x}_k)||^2 + ||\bar{x}_k - x^*||^2$$
$$+ \frac{1}{n}\sum_{i\in[n]}\left(||\bar{y}_k - y_{k,i}||^2 + ||\bar{x}_k - x_{k,i}||^2\right)] \leq \mathcal{O}\left(\frac{1}{T} + \frac{K^2}{T^2}\right), \tag{43}$$

*for some sequence of positive weights $w_k = k + \kappa$ where $\kappa \geq 4K$ and $W_T = \sum_{k=0}^{T}w_k$. As a consequence, it holds for any $i \in [n]$ that $\lim_{k\to\infty}||\bar{y}_k - y^*(\bar{x}_k)||^2 = 0$ a.s., $\lim_{k\to\infty}||\bar{x}_k - x^*||^2 = 0$ a.s., $\lim_{k\to\infty}||\bar{y}_k - y_{k,i}||^2 = 0$ a.s., and $\lim_{k\to\infty}||\bar{x}_k - x_{k,i}||^2 = 0$ a.s.*

**Remark 4.** *We see that $K$ only appears in the higher order term of (43), and the local TTSA with periodic global averaging converges at a tight rate $\mathcal{O}(1/T)$ when $T$ is sufficiently large. That is, the effects of multiple local updates become negligible after a few iterations, meaning that our algorithm gains communication efficiency through via infrequent communication, essentially for free.*

**Remark 5.** *We can extend Theorem 2 to the SBO and SCO problems. Consider the SBO algorithm with the updates in (13)-(14) and the SCO algorithm with the updates in (19)-(20). Assuming Assumptions 1-4 holds and selecting step sizes $\alpha_k = \Theta(\frac{1}{k+K})$ and $\beta_k = \Theta(\frac{1}{k+K})$, the resulting local SBO and SCO with periodic global averaging can converge at rate $\mathcal{O}\left(\frac{1}{T} + \frac{K^2}{T^2}\right)$.*

## 6  TTSA with Delayed Updates

In this section, we propose the third instance of EF-TTSA, namely, TTSA with delayed updates. First, we use the EF-TTSA framework to analyze TTSA with delayed updates, and then extend the results to the SBO and SCO problems.

---

**Algorithm 3** TTSA with Delayed Updates

---

1: **Initialization:** $\{\alpha_k\}_{k\geq 0}, \{\beta_k\}_{k\geq 0}, x_0 \in \mathbb{R}^{d_0}, y_0 \in \mathbb{R}^{d_1}$
2: **for** $k = 0, 1, \cdots, T-1$ **do**
3: $\quad x_{k+1} = x_k + \alpha_{k-\tau}f_{k-\tau}$
4: $\quad y_{k+1} = y_k + \beta_{k-\tau}g_{k-\tau}$
5: **end for**

---

The TTSA with delayed updates is illustrated in Algorithm 3. For a fixed (integer) delay $\tau \geq 1$, the sequence $\{x_k, y_k\}_{k\geq 0}$ is updated in the following way:

$$x_{k+1} = x_k + \alpha_{k-\tau}f_{k-\tau}, \tag{44}$$
$$y_{k+1} = y_k + \beta_{k-\tau}g_{k-\tau}, \tag{45}$$

where $f_{k-\tau} = f(x_{k-\tau}, y_{k-\tau}) + \xi_{k-\tau}$, and $g_{k-\tau} = g(x_{k-\tau}, y_{k-\tau}) + \psi_{k-\tau}$. Throughout this section, we use the convention that $f_{k-\tau} = g_{k-\tau} = 0$, if $k < \tau$. The delay may come from asynchrony in the development of distributed parallel algorithms; see e.g., Agarwal & Duchi (2011).

Observe that Algorithm 3 takes the following form in the EF-TTSA framework:

$$d_k = \sum_{i=1}^{\tau} \alpha_{k-i} f_{k-i}, \qquad \mu_k = \begin{cases} \alpha_{k-\tau} f_{k-\tau} & \text{if } k \geq \tau, \\ 0 & \text{if } k < \tau, \end{cases} \tag{46}$$

$$e_k = \sum_{i=1}^{\tau} \beta_{k-i} g_{k-i}, \qquad \nu_k = \begin{cases} \beta_{k-\tau} g_{k-\tau} & \text{if } k \geq \tau, \\ 0 & \text{if } k < \tau. \end{cases} \tag{47}$$

In view of (46)-(47), applying Lemma 1, we have

$$\Xi_{k+1} \leq (1 - \frac{\lambda_f}{2} \alpha_k)\Xi_k + \Delta_1 \alpha_k \Phi_k + \Delta_2 \alpha_k^2 (\sigma_\xi^2 + \sigma_\psi^2), \tag{48}$$

where $\Phi_k = \mathbb{E}\left[||d_k||^2 + ||e_k||^2\right]$. We present an upper bound on $\Phi_k$ in the following Lemma.

**Lemma 4.** *Let $\{x_k, d_k, y_k, e_k\}$ be the sequence generated by Algorithm 3. Suppose Assumptions 1-4 hold, and set $\alpha_k \leq \frac{1}{\sqrt{(4L_f^2 + 4L^2 + 3L_g^2 \bar{\beta}^2)\tau}}$ and $\beta_k = \bar{\beta} \alpha_k$, we have*

$$\Phi_k \leq 2(4L_f^2 + 3L_g^2 \bar{\beta}^2)\tau \sum_{j=k-\tau}^{k-1} \alpha_j^2 \Xi_j + 2(1 + \bar{\beta}^2) \sum_{j=k-\tau}^{k-1} \alpha_j^2 \left(\sigma_\xi^2 + \sigma_\psi^2\right). \tag{49}$$

Noting this fact, we provide the main theorem for TTSA with delayed updates in the next result.

**Theorem 3.** *Consider the sequence $\{x_k, y_k\}$ generated by Algorithm 3. Suppose Assumptions 1-4 hold. Selecting step sizes $\alpha_k = \Theta(\frac{1}{k+\tau})$ and $\beta_k = \Theta(\frac{1}{k+\tau})$, then it holds that*

$$\frac{1}{W_T} \sum_{k=0}^{T-1} w_k \mathbb{E}[||y_k - y^*(x_k)||^2 + ||x_k - x^*||^2] \leq \mathcal{O}\left(\frac{1}{T} + \frac{\tau^2}{T^2}\right), \tag{50}$$

*for some sequence of positive weights $w_k = k + \kappa$ where $\kappa \geq 4\tau$ and $W_T = \sum_{k=0}^{T} w_k$. As a consequence, it holds that $\lim_{k \to \infty} ||y_k - y^*(x_k)||^2 = 0$ a.s. and $\lim_{k \to \infty} ||x_k - x^*||^2 = 0$ a.s.*

**Remark 6.** *In analogy to Theorems 1 and 2, this result shows that the dominating term in the rate is not affected by the $\tau$ parameter. Moreover, we notice that the impact of the delay becomes negligible if $T = \Omega(\tau^2)$. The performance of TTSA with delays is comparable to that of TTSA without delays. We here focus on the fixed delay $\tau$. Indeed, our analysis applies to more general settings, see e.g., Feyzmahdavian et al. (2016); arbitrary delays upper bounded by $\tau$.*

**Remark 7.** *We can extend Theorem 3 to the SBO and SCO problems. Consider the SBO algorithm with the updates in (13)-(14) and the SCO algorithm with the updates in (19)-(20). Assuming Assumptions 1-4 holds and selecting step sizes $\alpha_k = \Theta(\frac{1}{k+\tau})$ and $\beta_k = \Theta(\frac{1}{k+\tau})$, the resulting SBO and SCO with delayed updates can converge at rate $\mathcal{O}\left(\frac{1}{T} + \frac{\tau^2}{T^2}\right)$.*

## 7 CONCLUSION AND FUTURE WORK

In this work, we consider error-feedback-based two-time-scale stochastic approximation, EF-TTSA, that captures a rich class of structured perturbations such as compression, local updates, and delays. We present a unified theory of EF-TTSA to analyze the impact of different forms of structured perturbations. This is (to the best of our knowledge) the first to demonstrate that two-time-scale stochastic approximation is robust to structured perturbations. In particular, we propose three instances of EF-TTSA, i.e., error-compensated TTSA with arbitrary compressors (Algorithm 1), local TTSA with periodic global averaging (Algorithm 2), and TTSA with delayed updates (Algorithm 3). We see that structured perturbations only affect the higher-order term of the convergence rate. That is, the effects of structured perturbations become negligible after a few iterations, and Algorithms 1, 2, and 3 converge at the same rate as standard TTSA without structured perturbations.

Future directions of this work include studying the EF-TTSA framework in the non-strongly monotone case and exploring multiple-time-scale stochastic approximation algorithms.

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
