APPENDIX

## A. MISSING PROOFS FOR EF-TTSA

### A1. INTERMEDIATE LEMMAS

We first present some intermediate lemmas, which shall be used to prove Lemma 1.

**Lemma 5.** Suppose Assumptions 1-4 hold. For any $k \geq 0$, we have

$$\mathbb{E}[||\bar{y}_{k+1} - \bar{y}_k^*||^2 | \mathcal{F}_k]$$
$$\leq (1 - \lambda_g \beta_k + 3\beta_k^2 L_g^2)||\bar{y}_k - \bar{y}_k^*||^2 + \beta_k^2 \sigma_\psi^2 + \left( \frac{2L_g^2}{\lambda_g} \beta_k + 3\beta_k^2 L_g^2 \right) (||d_k||^2 + ||e_k||^2). \quad (1)$$

*Proof of Lemma 5.* Observe that by the definition of $\bar{y}_k$, we have $\bar{y}_{k+1} = \bar{y}_k + \beta_k g_k$, implying that

$$\mathbb{E}[||\bar{y}_{k+1} - \bar{y}_k^*||^2 | \mathcal{F}_k] = ||\bar{y}_k - \bar{y}_k^*||^2 + 2\beta_k \mathbb{E}[\langle \bar{y}_k - \bar{y}_k^*, g_k \rangle | \mathcal{F}_k] + \beta_k^2 \mathbb{E}[||g_k||^2 | \mathcal{F}_k]. \quad (2)$$

Using the fact that $\langle x, y \rangle \leq \frac{\lambda_g}{4} x^2 + \frac{1}{\lambda_g} y^2$, we have

$$\mathbb{E}[\langle \bar{y}_k - \bar{y}_k^*, g_k \rangle | \mathcal{F}_k] = \langle \bar{y}_k - \bar{y}_k^*, g(\bar{x}_k, \bar{y}_k) \rangle + \langle \bar{y}_k - \bar{y}_k^*, g(\bar{x}_k, y_k) - g(\bar{x}_k, \bar{y}_k) \rangle$$
$$+ \langle \bar{y}_k - \bar{y}_k^*, g(x_k, y_k) - g(\bar{x}_k, y_k) \rangle$$
$$\leq -\frac{\lambda_g}{2} ||\bar{y}_k - \bar{y}_k^*||^2 + \frac{L_g^2}{\lambda_g} ||d_k||^2 + \frac{L_g^2}{\lambda_g} ||e_k||^2, \quad (3)$$

where the inequality holds due to Assumptions 2, 3 and the definitions of $d_k$ and $e_k$.

By Assumption 2, we obtain

$$\mathbb{E}[||g_k||^2 | \mathcal{F}_k] = \mathbb{E}[||g(x_k, y_k) + \psi_k||^2 | \mathcal{F}_k]$$
$$\leq \mathbb{E}[||g(x_k, y_k) - g(\bar{x}_k, y_k) + g(\bar{x}_k, y_k) - g(\bar{x}_k, \bar{y}_k) + g(\bar{x}_k, \bar{y}_k) - g(\bar{x}_k, \bar{y}_k^*)||^2 | \mathcal{F}_k] + \sigma_\psi^2$$
$$\leq 3L_g^2 ||\bar{y}_k - \bar{y}_k^*||^2 + 3L_g^2 ||d_k||^2 + 3L_g^2 ||e_k||^2 + \sigma_\psi^2. \quad (4)$$

Substituting the results from (3) and (4) into (2) completes the proof of Lemma 5. $\square$

**Lemma 6.** Suppose Assumptions 1-4 hold. For any $k \geq 0$, we have

$$\mathbb{E}[||\bar{y}_{k+1} - \bar{y}_{k+1}^*||^2 | \mathcal{F}_k] \leq (1 + L_{y,0} L_f \alpha_k + L_{y,0} L \alpha_k + 2\alpha_k^2 \sigma_\xi^2 L_{y,1} + \frac{4\alpha_k L_{y,0}^2 L^2}{\lambda_f}) \mathbb{E}[||\bar{y}_{k+1} - \bar{y}_k^*||^2 | \mathcal{F}_k]$$
$$+ (3\alpha_k L_{y,0} L_f + 4\alpha_k^2 L_{y,1} L_f^2 + 4\alpha_k^2 L_{y,0}^2 L_f^2)||\bar{y}_k - \bar{y}_k^*||^2$$
$$+ (\frac{\alpha_k \lambda_f}{4} + 4\alpha_k^2 L_{y,1} L^2 + 4\alpha_k^2 L_{y,0}^2 L^2)||\bar{x}_k - x^*||^2$$
$$+ (\alpha_k L_{y,0} L + 3\alpha_k L_{y,0}^3 L_f + 4\alpha_k^2 L_{y,1} L_f^2 + 4\alpha_k^2 L_{y,0}^2 L_f^2) ||d_k||^2$$
$$+ (3\alpha_k L_{y,0} L_f + 4\alpha_k^2 L_{y,1} L_f^2 + 4\alpha_k^2 L_{y,0}^2 L_f^2)||e_k||^2$$
$$+ \alpha_k^2 L_{y,1} \sigma_\xi^2 + \alpha_k^2 L_{y,0}^2 \sigma_\xi^2.$$

*Proof of Lemma 6.* For any $k \geq 0$, we have

$$\mathbb{E}[||\bar{y}_{k+1} - \bar{y}_{k+1}^*||^2 | \mathcal{F}_k] = \mathbb{E}[||\bar{y}_{k+1} - \bar{y}_k^*||^2 | \mathcal{F}_k] + 2\mathbb{E}[\langle \bar{y}_{k+1} - \bar{y}_k^*, \bar{y}_k^* - \bar{y}_{k+1}^* \rangle | \mathcal{F}_k]$$
$$+ \mathbb{E}[||\bar{y}_k^* - \bar{y}_{k+1}^*||^2 | \mathcal{F}_k]. \quad (5)$$

By the mean-value theorem, there exists $\widetilde{x}_{k+1} = a\bar{x}_k + (1-a)\bar{x}_{k+1}$ for $a \in [0,1]$ such that

$$
\begin{aligned}
\mathbb{E}[\langle \bar{y}_{k+1} - \bar{y}_k^*, \bar{y}_k^* - \bar{y}_{k+1}^* \rangle | \mathcal{F}_k] &= \mathbb{E}[\langle \bar{y}_k^* - \bar{y}_{k+1}, \nabla y^*(\widetilde{x}_{k+1})(\bar{x}_{k+1} - \bar{x}_k)\rangle | \mathcal{F}_k]\\
&= \alpha_k \mathbb{E}[\langle \bar{y}_k^* - \bar{y}_{k+1}, \nabla y^*(\widetilde{x}_{k+1})f_k\rangle | \mathcal{F}_k]\\
&= \alpha_k \underbrace{\mathbb{E}[\langle \bar{y}_k^* - \bar{y}_{k+1}, \nabla y^*(\widetilde{x}_{k+1})\xi_k\rangle | \mathcal{F}_k]}_{I_1}\\
&+ \alpha_k \underbrace{\mathbb{E}[\langle \bar{y}_k^* - \bar{y}_{k+1}, \nabla y^*(\widetilde{x}_{k+1})(f(x_k,y_k) - f(x_k, y_k^*))\rangle | \mathcal{F}_k]}_{I_2}\\
&+ \alpha_k \underbrace{\mathbb{E}[\langle \bar{y}_k^* - \bar{y}_{k+1}, \nabla y^*(\widetilde{x}_{k+1})(f(x_k, y_k^*) - f(x^*, y^*(x^*)))\rangle | \mathcal{F}_k]}_{I_3}. \quad (6)
\end{aligned}
$$

For the term $I_1$ in the above equation (6), it can be bounded as

$$
\begin{aligned}
I_1 &= \mathbb{E}[\langle \bar{y}_k^* - \bar{y}_{k+1}, (\nabla y^*(\widetilde{x}_{k+1}) - \nabla y^*(\bar{x}_k))\xi_k\rangle + \langle \bar{y}_k^* - \bar{y}_{k+1}, \nabla y^*(\bar{x}_k)\xi_k\rangle | \mathcal{F}_k].\\
&\leq \alpha_k L_{y,1}\mathbb{E}[||\bar{y}_k^* - \bar{y}_{k+1}|| \cdot ||f(x_k, y_k)|| \cdot ||\xi_k|| + ||\bar{y}_k^* - \bar{y}_{k+1}|| \cdot ||\xi_k|| \cdot ||\xi_k|| | \mathcal{F}_k]\\
&\leq \alpha_k L_{y,1}\sigma_\xi^2 \mathbb{E}[||\bar{y}_k^* - \bar{y}_{k+1}||^2 | \mathcal{F}_k] + \frac{1}{2}\alpha_k L_{y,1}||f(x_k, y_k)||^2 + \frac{1}{2}\alpha_k L_{y,1}\sigma_\xi^2\\
&\leq \alpha_k L_{y,1}\sigma_\xi^2 \mathbb{E}[||\bar{y}_k^* - \bar{y}_{k+1}||^2 | \mathcal{F}_k] + 2\alpha_k L_f^2 L_{y,1}||\bar{y}_k - \bar{y}_k^*||^2 + 2\alpha_k L^2 L_{y,1}||\bar{x}_k - x^*||^2\\
&\quad + 2\alpha_k L_f^2 L_{y,1}||d_k||^2 + 2\alpha_k L_f^2 L_{y,1}||e_k||^2 + \frac{1}{2}\alpha_k L_{y,1}\sigma_\xi^2. \quad (7)
\end{aligned}
$$

For the term $I_2$ in the above equation (6), it can be bounded as

$$
\begin{aligned}
I_2 &\leq L_{y,0}L_f \mathbb{E}[||\bar{y}_{k+1} - \bar{y}_k^*|| \cdot ||y_k - y_k^*|| | \mathcal{F}_k]\\
&\leq \frac{L_{y,0}L_f}{2}\mathbb{E}[||\bar{y}_{k+1} - \bar{y}_k^*||^2 | \mathcal{F}_k] + \frac{3L_{y,0}L_f}{2}||\bar{y}_k - \bar{y}_k^*||^2 + \frac{3L_{y,0}L_f}{2}||e_k||^2 + \frac{3L_{y,0}^3 L_f}{2}||d_k||^2.
\end{aligned}
$$
$$(8)$$

For the term $I_3$ in the above equation (6), it can be bounded as

$$
\begin{aligned}
I_3 &= L_{y,0}L\mathbb{E}[||\bar{y}_{k+1} - y^*(\bar{x}_k)|| \cdot (||\bar{x}_k - x_k|| + ||\bar{x}_k - x^*||) | \mathcal{F}_k]\\
&\leq (\frac{L_{y,0}L}{2} + \frac{2L_{y,0}^2 L^2}{\lambda_f})\mathbb{E}[||\bar{y}_{k+1} - y^*(\bar{x}_k)||^2 | \mathcal{F}_k] + \frac{L_{y,0}L}{2}||d_k||^2 + \frac{\lambda_f}{8}||\bar{x}_k - x^*||^2. \quad (9)
\end{aligned}
$$

In summary, substituting the results from (9), (8) and (7) into (6) gives

$$
\begin{aligned}
\mathbb{E}&\left[\langle \bar{y}_{k+1} - y^*(\bar{x}_k), y^*(\bar{x}_k) - y^*(\bar{x}_{k+1})\rangle | \mathcal{F}_k\right]\\
&\leq (\frac{L_{y,0}L_f\alpha_k}{2} + \frac{L_{y,0}L\alpha_k}{2} + \frac{2\alpha_k L_{y,0}^2 L^2}{\lambda_f} + \alpha_k^2 L_{y,1}\sigma_\xi^2)\mathbb{E}\left[||\bar{y}_{k+1} - \bar{y}_k^*||^2 | \mathcal{F}_k\right] + \frac{1}{2}\alpha_k^2 L_{y,1}\sigma_\xi^2\\
&\quad + (\frac{3\alpha_k L_{y,0}L_f}{2} + 2\alpha_k^2 L_{y,1}L_f^2)||\bar{y}_k - \bar{y}_k^*||^2 + (\frac{\alpha_k \lambda_f}{8} + 2\alpha_k^2 L_{y,1}L^2)||\bar{x}_k - x^*||^2\\
&\quad + (\frac{\alpha_k L_{y,0}L}{2} + \frac{3\alpha_k L_{y,0}^3 L_f}{2} + 2\alpha_k^2 L_{y,1}L_f^2)||d_k||^2\\
&\quad + (\frac{3\alpha_k L_{y,0}L_f}{2} + 2\alpha_k^2 L_{y,1}L_f^2)||e_k||^2. \quad (10)
\end{aligned}
$$

We observe that the last term in the equation (5) can be bounded as

$$
\begin{aligned}
\mathbb{E}[||\bar{y}_k^* - \bar{y}_{k+1}^*||^2 | \mathcal{F}_k] &\leq \alpha_k^2 L_{y,0}^2 \mathbb{E}[||f_k||^2 | \mathcal{F}_k]\\
&\leq 4\alpha_k^2 L_{y,0}^2 L_f^2||\bar{y}_k - \bar{y}_k^*||^2 + 4\alpha_k^2 L_{y,0}^2 L^2||\bar{x}_k - x^*||^2 + \alpha_k^2 L_{y,0}^2 \sigma_\xi^2\\
&\quad + 4\alpha_k^2 L_{y,0}^2 L_f^2\left(||d_k||^2 + ||e_k||^2\right). \quad (11)
\end{aligned}
$$

Substituting the results from (10) and (11) into (5) completes the proof of Lemma 6. $\qquad\square$

**Lemma 7.** Suppose Assumptions 1-4 hold. For any $k \geq 0$, we have

$$\mathbb{E}[||\bar{x}_{k+1} - x^*||^2 | \mathcal{F}_k] \leq (1 - \lambda_f \alpha_k + 4\alpha_k^2 L^2)||\bar{x}_k - x^*||^2$$
$$+ (\frac{9L_f^2}{\lambda_f}\alpha_k + 4\alpha_k^2 L_f^2)||\bar{y}_k - \bar{y}_k^*||^2 + \alpha_k^2 \sigma_\xi^2$$
$$+ (\frac{3L_f^2}{\lambda_f}\alpha_k + \frac{12L_f^2 L_{y,0}^2}{\lambda_f}\alpha_k + 4\alpha_k^2 L_f^2)||d_k||^2 + (\frac{9L_f^2}{\lambda_f}\alpha_k + 4\alpha_k^2 L_f^2)||e_k||^2. \quad (12)$$

*Proof of Lemma 7.* Observe that by the definition of $\bar{x}_k$, we have $\bar{x}_{k+1} = \bar{x}_k + \alpha_k f_k$, implying that

$$\mathbb{E}[||\bar{x}_{k+1} - x^*||^2 | \mathcal{F}_k] = ||\bar{x}_k - x^*||^2 + 2\alpha_k \mathbb{E}[\langle \bar{x}_k - x^*, f_k \rangle | \mathcal{F}_k] + \alpha_k^2 \mathbb{E}[||f_k||^2 | \mathcal{F}_k]. \quad (13)$$

Using the fact that $\langle x, y \rangle \leq \frac{\lambda_f}{6} x^2 + \frac{3}{2\lambda_f} y^2$, we have

$$\mathbb{E}[\langle \bar{x}_k - x^*, f_k \rangle | \mathcal{F}_k] = \langle \bar{x}_k - x^*, f(\bar{x}_k, \bar{y}_k^*) \rangle + \langle \bar{x}_k - x^*, f(\bar{x}_k, y_k^*) - f(\bar{x}_k, \bar{y}_k^*) \rangle$$
$$+ \langle \bar{x}_k - x^*, f(\bar{x}_k, y_k) - f(\bar{x}_k, y_k^*) \rangle + \langle \bar{x}_k - x^*, f(x_k, y_k) - f(\bar{x}_k, y_k) \rangle$$
$$\leq -\frac{\lambda_f}{2}||\bar{x}_k - x^*||^2 + \frac{3L_{y,0}^2 L_f^2}{2\lambda_f}||\bar{x}_k - x_k||^2 + \frac{3L_f^2}{2\lambda_f}||y_k - y_k^*||^2 + \frac{3L_f^2}{2\lambda_f}||\bar{x}_k - x_k||^2$$
$$\leq -\frac{\lambda_f}{2}||\bar{x}_k - x^*||^2 + \frac{9L_f^2}{2\lambda_f}||\bar{y}_k - \bar{y}_k^*||^2 + \frac{12L_f^2 L_{y,0}^2 + 3L_f^2}{2\lambda_f}||d_k||^2 + \frac{9L_f^2}{2\lambda_f}||e_k||^2, \quad (14)$$

where the inequality is by Assumptions 1, 2, 3, and the definitions of $d_k$ and $e_k$.

By Assumptions 2 and 4 we obtain

$$\mathbb{E}[||f_k||^2 | \mathcal{F}_k] \leq 4L_f^2 ||\bar{y}_k - \bar{y}_k^*||^2 + 4L^2 ||\bar{x}_k - x^*||^2 + 4L_f^2(||d_k||^2 + ||e_k||^2) + \sigma_\xi^2. \quad (15)$$

Substituting the results from (14) and (15) into (13) completes the proof of Lemma 7. $\qquad \square$

## A2. Proof of Lemma 1

*Proof of Lemma 1.* Noting that by Lemmas 5, 6, 7, and with the choice of $\beta_k = \frac{2c_1 + \lambda_f}{\lambda_g}\alpha_k$, we have

$$\mathbb{E}[||\bar{y}_{k+1} - \bar{y}_{k+1}^*||^2 + ||\bar{x}_{k+1} - x^*||^2 | \mathcal{F}_k] \leq \left( (1 + c_2 \alpha_k)(1 - \lambda_g \beta_k + 3\beta_k^2 L_g^2) + c_4 \alpha_k \right) ||\bar{y}_k - \bar{y}_k^*||^2$$
$$+ \left( 1 - \frac{3}{4}\alpha_k \lambda_f + 4\alpha_k^2 L_y L^2 \right) ||\bar{x}_k - x^*||^2$$
$$+ \Delta_1 \alpha_k(||d_k||^2 + ||e_k||^2) + \Delta_2 \alpha_k^2(\sigma_\xi^2 + \sigma_\psi^2), \quad (16)$$

where $\Delta_1 = (1 + c_2)\left(3L_g^2 + 2L_g^2/\lambda_g\right)\bar{\beta} + c_3$, $\Delta_2 = (1 + c_2)\bar{\beta}^2 + L_y$, $c_1 = L_{y,0}L + 4L_{y,0}L_f + 2L_{y,1}\sigma_\xi^2 + 4L_y L_f^2 + \frac{4L_{y,0}^2 L^2 + 9L_f^2}{\lambda_f}$, $c_2 = L_{y,0}L_f + L_{y,0}L + 2L_{y,1}\sigma_\xi^2 + \frac{4L_{y,0}^2 L^2}{\lambda_f}$, $c_3 = L_{y,0}L + 3L_{y,0}L_y L_f + 4L_y L_f^2 + \frac{12L_f^2 L_y}{\lambda_f}$, $c_4 = 3L_{y,0}L_f + 4L_y L_f^2 + \frac{9L_f^2}{\lambda_f}$, and $L_y = L_{y,0}^2 + L_{y,1} + 1$. Moreover, with the choice of $\alpha_k \leq \min\{\frac{\lambda_f}{16L_y L^2}, \frac{\lambda_g^2}{6L_g^2(2c_1 + \lambda_f)}\}$, we have

$$(1 + c_2 \alpha_k)(1 - \lambda_g \beta_k + 3\beta_k^2 L_g^2) + c_4 \alpha_k \leq 1 - \frac{\lambda_f}{2}\alpha_k. \quad (17)$$

which together with (16) completes the proof of Lemma 1. $\qquad \square$

# B. Missing Proofs for Algorithm 1

## B.1. Proof of Lemma 2

We first prove lemma 2, which shall be used to prove Theorem 1.

*Proof of Lemma 2.* By Assumption 5, for any $k \geq 0$, we have

$$\mathbb{E}[||d_{k+1,i}||^2|\mathcal{F}_k] = \mathbb{E}[||d_{k,i} + \alpha_k f_{k,i} - \mathcal{Q}[d_{k,i} + \alpha_k f_{k,i}]||^2|\mathcal{F}_k]$$
$$\leq (1-\delta)\mathbb{E}[||d_{k,i} + \alpha_k f_{k,i}||^2|\mathcal{F}_k]$$
$$\leq (1-\delta)(1+a)||d_{k,i}||^2 + (1-\delta)(1+a^{-1})\alpha_k^2||f(x_k.y_k)||^2 + (1-\delta)\alpha_k^2\sigma_\xi^2, \quad (18)$$

where the last inequality holds since $||x+y||^2 \leq (1+a)||x||^2 + (1+a^{-1})||y||^2$ for any $a > 0$.
By Assumptions 1 and 2 we obtain

$$||f(x_k, y_k)||^2 = ||f(x_k, y_k) - f(\bar{x}_k, y_k) + f(\bar{x}_k, y_k) - f(\bar{x}_k, \bar{y}_k)$$
$$+ f(\bar{x}_k, \bar{y}_k) - f(\bar{x}_k, \bar{y}_k^*) + f(\bar{x}_k, \bar{y}_k^*) - f(x^*, y^*(x^*))||^2$$
$$\leq 4L_f^2||\bar{y}_k - \bar{y}_k^*||^2 + 4L^2||\bar{x}_k - x^*||^2 + \frac{4L_f^2}{n}\sum_{i\in[n]}(||d_{k,i}||^2 + ||e_{k,i}||^2). \quad (19)$$

Substituting the results from (19) into (18) we obtain

$$\frac{1}{n}\sum_{i\in[n]}\mathbb{E}[||d_{k+1,i}||^2|\mathcal{F}_k] \leq \left((1-\delta)(1+a) + (1-\delta)(1+a^{-1})4\alpha_k^2 L_f^2\right)\frac{1}{n}\sum_{i\in[n]}||d_{k,i}||^2$$
$$+ (1-\delta)(1+a^{-1})4\alpha_k^2 L_f^2\frac{1}{n}\sum_{i\in[n]}||e_{k,i}||^2 + (1-\delta)\alpha_k^2\sigma_\xi^2$$
$$+ (1-\delta)(1+a^{-1})4\alpha_k^2 L_f^2||\bar{y}_k - \bar{y}_k^*||^2 + (1-\delta)(1+a^{-1})4\alpha_k^2 L^2||\bar{x}_k - x^*||^2. \quad (20)$$

Similarly, by Assumption 5, for any $k \geq 0$, we have

$$\mathbb{E}[||e_{k+1,i}||^2|\mathcal{F}_k] = \mathbb{E}[||e_{k,i} + \beta_k g_{k,i} - \mathcal{Q}[e_{k,i} + \beta_k g_{k,i}]||^2|\mathcal{F}_k]$$
$$\leq (1-\delta)\mathbb{E}[||e_{k,i} + \beta_k g_{k,i}||^2|\mathcal{F}_k]$$
$$\leq (1-\delta)(1+b)||e_{k,i}||^2 + (1-\delta)(1+b^{-1})\beta_k^2||g(x_k,y_k)||^2 + (1-\delta)\beta_k^2\sigma_\psi^2, \quad (21)$$

where the last inequality holds since $||x+y||^2 \leq (1+b)||x||^2 + (1+b^{-1})||y||^2$ for any $b > 0$.
By Assumption 1, we obtain

$$||g(x_k, y_k)||^2 \leq ||g(x_k, y_k) - g(\bar{x}_k, y_k) + g(\bar{x}_k, y_k) - g(\bar{x}_k, \bar{y}_k) + g(\bar{x}_k, \bar{y}_k) - g(\bar{x}_k, \bar{y}_k^*)||^2$$
$$\leq 3L_g^2||\bar{y}_k - \bar{y}_k^*||^2 + 3L_g^2\frac{1}{n}\sum_{i\in[n]}||d_{k,i}||^2 + 3L_g^2\frac{1}{n}\sum_{i\in[n]}||e_{k,i}||^2. \quad (22)$$

Substituting the results from (22) into (21) yields

$$\frac{1}{n}\sum_{i\in[n]}\mathbb{E}[||e_{k+1,i}||^2|\mathcal{F}_k] \leq \left((1-\delta)(1+b) + (1-\delta)(1+b^{-1})3\beta_k^2 L_g^2\right)\frac{1}{n}\sum_{i\in[n]}||e_{k,i}||^2$$
$$+ (1-\delta)(1+b^{-1})3\beta_k^2 L_g^2\frac{1}{n}\sum_{i\in[n]}||d_{k,i}||^2 + (1-\delta)\beta_k^2\sigma_\psi^2$$
$$+ \left((1-\delta)(1+b^{-1})3\beta_k^2 L_g^2\right)||\bar{y}_k - \bar{y}_k^*||^2. \quad (23)$$

Letting $a = b = \frac{\delta}{4}$. Combining the results from (20) and (23) we obtain

$$\frac{1}{n}\sum_{i\in[n]}\mathbb{E}[||d_{k+1,i}||^2 + ||e_{k+1,i}||^2|\mathcal{F}_k] \leq (1-\frac{\delta}{2})\frac{1}{n}\sum_{i\in[n]}\left(||d_{k,i}||^2 + ||e_{k,i}||^2\right) \quad (24)$$

$$+ (1-\delta)\left(1+\bar{\beta}^2\right)\alpha_k^2(\sigma_\xi^2 + \sigma_\psi^2) \quad (25)$$

$$+ (1-\delta)(1+\frac{4}{\delta})\left(4L_f^2 + 3\bar{\beta}^2 L_g^2\right)\alpha_k^2||\bar{y}_k - \bar{y}_k^*||^2 \quad (26)$$

$$+ (1-\delta)\left(1+\frac{4}{\delta}\right)4\alpha_k^2 L^2||\bar{x}_k - x^*||^2, \quad (27)$$

where the inequality is by the choice of $\{\alpha_k\}$ and $\{\beta_k\}$. This completes the proof of Lemma 2. $\quad\square$

## B.2. PROOF OF THEOREM 1

*Proof of Theorem 1.* By Lemma 2, we have

$$\Phi_{k+1} \leq \left(1 - \frac{\delta}{2}\right)\Phi_k + \frac{\Delta_3}{\delta}\alpha_k^2\Xi_k + \Delta_4\alpha_k^2\left(\sigma_\xi^2 + \sigma_\psi^2\right), \tag{28}$$

where $\Delta_3 = 5(4L_f^2 + 4L^2 + 3L_g^2\bar{\beta}^2)$ and $\Delta_4 = 1 + \bar{\beta}^2$. Unrolling $\Phi_k$ recursively up to 0 we get

$$\Phi_{k+1} \leq \frac{\Delta_3}{\delta}\sum_{t=0}^{k}(1 - \frac{\delta}{2})^{k-t}\alpha_t^2\Xi_t + \Delta_4\sum_{t=0}^{k}(1 - \frac{\delta}{2})^{k-t}\alpha_t^2\left(\sigma_\xi^2 + \sigma_\psi^2\right). \tag{29}$$

Let $\alpha_k = \frac{8}{\lambda_f(\kappa+k)}$ and $w_k = \kappa + k$ where $\kappa \geq \frac{16}{\delta}$. We see that $w_{k+1} \leq w_k(1 + \frac{\delta}{4})$. Multiplying both sides of (29) by $w_{k+1}$, we obtain

$$w_{k+1}\Phi_{k+1} \leq \frac{\Delta_3}{\delta}(1 + \frac{\delta}{4})\sum_{t=0}^{k}(1 - \frac{\delta}{2})^{k-t}\alpha_t^2\Xi_t + \Delta_4(1 + \frac{\delta}{4})\sum_{t=0}^{k}(1 - \frac{\delta}{2})^{k-t}\alpha_t^2\left(\sigma_\xi^2 + \sigma_\psi^2\right). \tag{30}$$

Summing (30) over $k = 0$ to $T - 1$ yields

$$\sum_{t=0}^{T-1}w_k\Phi_k \leq \frac{\Delta_3}{\delta}(1 + \frac{\delta}{4})\sum_{k=0}^{T-1}w_k\sum_{t=0}^{k}(1 - \frac{\delta}{2})^{k-t}\alpha_t^2\Xi_t$$

$$+ \Delta_4(1 + \frac{\delta}{4})\sum_{k=0}^{T-1}w_k\sum_{t=0}^{k}(1 - \frac{\delta}{2})^{k-t}\alpha_t^2(\sigma_\xi^2 + \sigma_\psi^2)$$

$$\leq \frac{\Delta_3}{\delta}(1 + \frac{\delta}{4})\sum_{k=0}^{T-1}\sum_{t=0}^{k}(1 - \frac{\delta}{4})^{k-t}\alpha_t^2 w_t\Xi_t$$

$$+ \Delta_4(1 + \frac{\delta}{4})\sum_{k=0}^{T-1}\sum_{t=0}^{k}(1 - \frac{\delta}{4})^{k-t}\alpha_t^2(\sigma_\xi^2 + \sigma_\psi^2)$$

$$\leq \frac{4\Delta_3}{\delta^2}(1 + \frac{\delta}{4})\sum_{k=0}^{T-1}\alpha_k^2 w_k\Xi_k + \frac{4\Delta_4}{\delta}(1 + \frac{\delta}{4})\sum_{k=0}^{T-1}\alpha_k^2(\sigma_\xi^2 + \sigma_\psi^2). \tag{31}$$

By Lemma 1, we have

$$\Xi_{k+1} \leq \left(1 - \frac{\lambda_f}{2}\alpha_k\right)\Xi_k + \Delta_1\alpha_k\Phi_k + \Delta_2\alpha_k^2(\sigma_\xi^2 + \sigma_\psi^2). \tag{32}$$

Multiplying both sides of (32) by $\frac{4w_k}{\lambda_f\alpha_k}$ and rearranging the terms, we obtain

$$\sum_{k=0}^{T-1}w_k\Xi_k \leq 4\sum_{k=0}^{T-1}\left(\left(1 - \frac{\lambda_f}{4}\alpha_k\right)\frac{w_k\Xi_k}{\lambda_f\alpha_k} - \frac{w_k\Xi_{k+1}}{\lambda_f\alpha_k}\right) + \frac{4\Delta_1}{\lambda_f}\sum_{k=0}^{T-1}w_k\Phi_k$$

$$+ \frac{4\Delta_2}{\lambda_f}(\sigma_\xi^2 + \sigma_\psi^2)\sum_{k=0}^{T-1}w_k\alpha_k. \tag{33}$$

Let $w_k = \kappa + k$ and $W_T = \sum_{k=0}^{T-1}w_k$. Observe that

$$(1 - \frac{\lambda_f\alpha_k}{4})\frac{w_k}{\lambda_f\alpha_k} = \frac{1}{8}(\kappa + k - 2)(\kappa + k) \leq \frac{1}{8}((\kappa + k - 1)^2 - 1) \leq \frac{1}{8}(\kappa + k - 1)^2. \tag{34}$$

Then, the above inequality (33) can be simplified as

$$\sum_{k=0}^{T-1}w_k\Xi_k \leq \frac{1}{2}\sum_{k=0}^{T-1}\left((\kappa + k - 1)^2\Xi_k - (\kappa + k)^2\Xi_{k+1}\right) + \frac{4\Delta_1}{\lambda_f}\sum_{k=0}^{T-1}w_k\Phi_k$$

$$+ \frac{4\Delta_2}{\lambda_f}(\sigma_\xi^2 + \sigma_\psi^2)\sum_{k=0}^{T-1}w_k\alpha_k$$

$$\leq \frac{1}{2}(\kappa - 1)^2\Xi_0^2 + \frac{4\Delta_1}{\lambda_f}\sum_{k=0}^{T-1}w_k\Phi_k + \frac{4\Delta_2}{\lambda_f}(\sigma_\xi^2 + \sigma_\psi^2)\sum_{k=0}^{T-1}w_k\alpha_k. \tag{35}$$

Substituting the results from (31) into (35) we obtain

$$\sum_{k=0}^{T-1} w_k (\Xi_k + \Phi_k) \le (\kappa - 1)^2 \Xi_0^2 + \frac{8\Delta_4}{\delta}(1 + \frac{4\Delta_1}{\lambda_f})(1 + \frac{\delta}{4}) \sum_{k=0}^{T-1} \alpha_k^2 w_k (\sigma_\xi^2 + \sigma_\psi^2)$$

$$+ \frac{8\Delta_2}{\lambda_f} \sum_{k=0}^{T-1} \alpha_k w_k (\sigma_\xi^2 + \sigma_\psi^2), \tag{36}$$

where the inequality is by the choice of step size satisfying $(1 + \frac{4\Delta_1}{\lambda_f})(1 + \frac{\delta}{4})\frac{4\Delta_3}{\delta^2}\alpha_k^2 \le \frac{1}{2}$. With $W_T = \sum_{k=0}^{T-1} w_k$, we have that $W_T = \sum_{k=0}^{T-1} w_k = \frac{(2\kappa + T - 1)T}{2} \ge \frac{T^2}{2}$.

Dividing both sides of (36) by $W_T$ and rearranging the terms, we obtain

$$\frac{1}{W_T} \sum_{k=0}^{T-1} w_k (\Xi_k + \Phi_k) \le \mathcal{O}\left(\frac{1}{T} + \frac{1}{\delta^2 T^2}\right), \tag{37}$$

where we omit the constants involved are only numerical constants that are independent of $\delta$ and $T$. Since $||y_k - y^*(x_k)||^2 + ||x_k - x^*||^2 \le \mathcal{O}(\Xi_k + \Phi_k)$, we prove the first part of Theorem 1. Then, Robbins-Siegmund's theorem which together with the choice of $\alpha_k = \Theta(\frac{1}{k+1/\delta})$ and $\beta_k = \Theta(\frac{1}{k+1/\delta})$ immediately implies that $\lim_{k\to\infty} ||y_k - y^*(x_k)||^2 = 0$ a.s., and $\lim_{k\to\infty} ||x_k - x^*||^2 = 0$ a.s. $\qquad\square$

## C. MISSING PROOFS FOR ALGORITHM 2

We first prove lemma 3, which shall be used to prove Theorem 2.

### C.1. PROOF OF LEMMA 3

*Proof of Lemma 3.* For every $k \ge 0$, denoting $k_0 = \lfloor k/K \rfloor$. By Algorithm 2, we have that $\bar{x}_{k_0} = x_{k_0,i}$ for $\forall i \in [n]$. Using the fact that $\mathbb{E}[||X - \mathbb{E}[X]||^2] = \mathbb{E}[||X||^2] - ||\mathbb{E}[X]||^2$, we have

$$\frac{1}{n}\sum_{i\in[n]} ||\bar{x}_k - x_{k,i}||^2 = \frac{1}{n}\sum_{i\in[n]} ||\bar{x}_k - \bar{x}_{k_0} + x_{k_0,i} - x_{k,i}||^2 \le \frac{1}{n}\sum_{i\in[n]} ||x_{k,i} - x_{k_0,i}||^2. \tag{38}$$

Observe that $x_{k,i} - x_{k_0,i} = \sum_{j=k_0}^{k-1} \alpha_j f_{j,i} = \sum_{j=k_0}^{k-1} \alpha_j (f(x_{j,i}, y_{j,i}) + \xi_{j,i})$. Then, we have

$$\frac{1}{n}\sum_{i\in[n]} ||\bar{x}_k - x_{k,i}||^2 \le \frac{1}{n}\sum_{i\in[n]} ||\sum_{j=k_0}^{k-1} \alpha_j (f(x_{j,i}, y_{j,i}) + \xi_{j,i})||^2$$

$$\le \frac{1}{n}\sum_{i\in[n]} K \sum_{j=k_0}^{k-1} ||\alpha_j f(x_{j,i}, y_{j,i})||^2 + \sum_{j=k_0}^{k-1} \alpha_j^2 \sigma_\xi^2. \tag{39}$$

By Assumption 2 we obtain

$$||f(x_{j,i}, y_{j,i})||^2 \le ||f(x_{j,i}, y_{j,i}) - f(\bar{x}_j, y_{j,i}) + f(\bar{x}_j, y_{j,i}) - f(\bar{x}_j, \bar{y}_j)$$

$$+ f(\bar{x}_j, \bar{y}_j) - f(\bar{x}_j, \bar{y}_j^*) + f(\bar{x}_j, \bar{y}_j^*) - f(x^*, y^*(x^*))||^2$$

$$\le 4L_f^2 ||\bar{y}_j - \bar{y}_j^*||^2 + 4L^2 ||\bar{x}_j - x^*||^2 + 4L_f^2 ||d_{j,i}||^2 + 4L_f^2 ||e_{j,i}||^2. \tag{40}$$

Substituting the results from (40) into (39) we obtain

$$\frac{1}{n}\sum_{i\in[n]} ||\bar{x}_k - x_{k,i}||^2 \le 4L_f^2 K \sum_{j=k_0}^{k-1} \alpha_j^2 ||\bar{y}_j - \bar{y}_j^*||^2 + 4L^2 K \sum_{j=k_0}^{k-1} \alpha_j^2 ||\bar{x}_j - x^*||^2$$

$$+ 4L_f^2 K \frac{1}{n}\sum_{i\in[n]}\sum_{j=k_0}^{k-1} \alpha_j^2 (||d_{j,i}||^2 + ||e_{j,i}||^2) + \sum_{j=k_0}^{k-1} \alpha_j^2 \sigma_\xi^2. \tag{41}$$

Similarly, using the fact that $\bar{y}_{k_0} = y_{k_0,i}$ for $\forall i \in [n]$, we have

$$\frac{1}{n} \sum_{i \in [n]} ||\bar{y}_k - y_{k,i}||^2 = \frac{1}{n} \sum_{i \in [n]} ||\bar{y}_k - \bar{y}_{k_0} + y_{k_0,i} - y_{k,i}||^2 \leq \frac{1}{n} \sum_{i \in [n]} ||y_{k,i} - y_{k_0,i}||^2. \tag{42}$$

and

$$y_{k,i} - y_{k_0,i} = \sum_{j=k_0}^{k-1} \beta_j g_{j,i} = \sum_{j=k_0}^{k-1} \beta_j \left( g(x_{j,i}, y_{j,i}) + \psi_{j,i} \right). \tag{43}$$

Then, we have

$$\frac{1}{n} \sum_{i \in [n]} ||\bar{y}_k - y_{k,i}||^2 \leq \frac{1}{n} \sum_{i \in [n]} K \sum_{j=k_0}^{k-1} ||\beta_j g(x_{j,i}, y_{j,i})||^2 + \sum_{j=k_0}^{k-1} \beta_j^2 \sigma_\psi^2. \tag{44}$$

By Assumption 2, we obtain

$$||g(x_{j,i}, y_{j,i})||^2 \leq ||g(x_{j,i}, y_{j,i}) - g(\bar{x}_j, y_{j,i}) + g(\bar{x}_j, y_{j,i}) - g(\bar{x}_j, \bar{y}_j) + g(\bar{x}_j, \bar{y}_j) - g(\bar{x}_j, \bar{y}_j^*)||^2$$

$$\leq 3L_g^2 ||\bar{y}_j - \bar{y}_j^*||^2 + 3L_g^2 \frac{1}{n} \sum_{i \in [n]} \left( ||d_{k,i}||^2 + ||e_{j,i}||^2 \right). \tag{45}$$

Substituting the results from (45) into (44) gives

$$\frac{1}{n} \sum_{i \in [n]} ||\bar{y}_k - y_{k,i}||^2 \leq 3L_g^2 K \sum_{j=k_0}^{k-1} \beta_j^2 ||\bar{y}_j - \bar{y}_j^*||^2$$

$$+ 3L_g^2 K \frac{1}{n} \sum_{i \in [n]} \sum_{j=k_0}^{k-1} \beta_j^2 \left( ||d_{j,i}||^2 + ||e_{j,i}||^2 \right) + \sum_{j=k_0}^{k-1} \beta_j^2 \sigma_\psi^2. \tag{46}$$

Combining the results from (41) and (46), and using the definitions of $\Xi_k$ and $\Phi_k$, we get

$$\Phi_k \leq (4L_f^2 + 4L^2 + 3L_g^2 \bar{\beta}^2) K \sum_{j=k_0}^{k-1} \alpha_j^2 \Phi_j + (4L_f^2 + 3L_g^2 \bar{\beta}^2) K \sum_{j=k_0}^{k-1} \alpha_j^2 \Xi_j$$

$$+ (1 + \bar{\beta}^2) \sum_{j=k_0}^{k-1} \alpha_j^2 (\sigma_\xi^2 + \sigma_\psi^2). \tag{47}$$

Recursively substituting every $\Phi_j$ for $j \geq k_0$ we obtain

$$\Phi_k \leq (4L_f^2 + 3L_g^2 \bar{\beta}^2) K \prod_{t=k_0+1}^{k-1} \left( 1 + (4L_f^2 + 4L^2 + 3L_g^2 \bar{\beta}^2) K \alpha_t^2 \right) \sum_{j=k_0}^{k-1} \alpha_j^2 \Xi_j$$

$$+ (1 + \bar{\beta}^2) \prod_{t=k_0+1}^{k-1} \left( 1 + (4L_f^2 + 4L^2 + 3L_g^2 \bar{\beta}^2) K \alpha_t^2 \right) \sum_{j=k_0}^{k-1} \alpha_j^2 \left( \sigma_\xi^2 + \sigma_\psi^2 \right). \tag{48}$$

Observe that

$$\prod_{t=k_0+1}^{k-1} \left( 1 + (4L_f^2 + 4L^2 + 3L_g^2 \bar{\beta}^2) K \alpha_t^2 \right) \leq \left( 1 + \frac{1}{K} \right)^K \leq 2, \tag{49}$$

where the inequality is by the choice of $\alpha_k \leq \frac{1}{\sqrt{(4L_f^2 + 4L^2 + 3L_g^2 \bar{\beta}^2) K}}$. Then, (48) can be simplified as

$$\Phi_k \leq (8L_f^2 + 6L_g^2 \bar{\beta}^2) K \sum_{j=k_0}^{k-1} \alpha_j^2 \Xi_j + 2(1 + \bar{\beta}^2) \sum_{j=k_0}^{k-1} \alpha_j^2 \left( \sigma_\xi^2 + \sigma_\psi^2 \right), \tag{50}$$

This completes the proof. $\qquad\qquad\square$

## C.2. PROOF OF THEOREM 2

*Proof of Theorem 2.* By Lemma 1, we have

$$\Xi_{k+1} \le \left(1 - \frac{\lambda_g}{2}\alpha_k\right)\Xi_k + \Delta_1\alpha_k\Phi_k + \Delta_2\alpha_k^2(\sigma_\xi^2 + \sigma_\psi^2). \tag{51}$$

Multiplying both sides of (51) by $\frac{4w_k}{\lambda_f\alpha_k}$ and rearranging the terms, we obtain

$$\sum_{k=0}^{T-1} w_k\Xi_k \le 4\sum_{k=0}^{T-1}\left((1 - \frac{\lambda_f}{4}\alpha_k)\frac{w_k\Xi_k}{\lambda_f\alpha_k} - \frac{w_k\Xi_{k+1}}{\lambda_f\alpha_k}\right) + \frac{4\Delta_1}{\lambda_f}\sum_{k=0}^{T-1} w_k\Phi_k$$
$$+ \frac{4\Delta_2}{\lambda_f}(\sigma_\xi^2 + \sigma_\psi^2)\sum_{k=0}^{T-1} w_k\alpha_k. \tag{52}$$

Let $\alpha_k = \frac{8}{\lambda_f(k+\kappa)}$ and $w_k = k + \kappa$ where $\kappa \ge 4K$ and $W_T = \sum_{k=0}^{T-1} w_k$. Observe that

$$(1 - \frac{\lambda_f\alpha_k}{4})\frac{w_k}{\lambda_f\alpha_k} = \frac{1}{8}\left(\kappa + k - 2\right)\left(\kappa + k\right) \le \frac{1}{8}((\kappa + k - 1)^2 - 1) \le \frac{1}{8}(\kappa + k - 1)^2. \tag{53}$$

Then, the above inequality (52) can be simplified as

$$\sum_{k=0}^{T-1} w_k\Xi_k \le \frac{1}{2}\sum_{k=0}^{T-1}\left((\kappa + k - 1)^2\Xi_k - (\kappa + k)^2\Xi_{k+1}\right) + \frac{4\Delta_1}{\lambda_f}\sum_{k=0}^{T-1} w_k\Phi_k$$
$$+ \frac{4\Delta_2}{\lambda_f}(\sigma_\xi^2 + \sigma_\psi^2)\sum_{k=0}^{T-1} w_k\alpha_k$$
$$\le \frac{1}{2}(\kappa - 1)^2\Xi_0^2 + \frac{4\Delta_1}{\lambda_f}\sum_{k=0}^{T-1} w_k\Phi_k + \frac{4\Delta_2}{\lambda_f}(\sigma_\xi^2 + \sigma_\psi^2)\sum_{k=0}^{T-1} w_k\alpha_k. \tag{54}$$

By Lemma 3, we have

$$\Phi_k \le \Delta_5 K\sum_{j=k_0}^{k-1}\alpha_j^2\Xi_j + \Delta_6\sum_{j=k_0}^{k-1}\alpha_j^2\left(\sigma_\xi^2 + \sigma_\psi^2\right), \tag{55}$$

where $\Delta_5 = 8L_f^2 + 6L_g^2\bar{\beta}^2$ and $\Delta_6 = 2(1 + \bar{\beta}^2)$. Summing (55) over $k = 0$ to $T - 1$ yields

$$\sum_{k=0}^{T-1} w_k\Phi_k \le \Delta_5 K\sum_{k=0}^{T-1}\sum_{j=k_0}^{k-1}(1 + \frac{1}{K})^{k-j}\alpha_j^2 w_j\Xi_j + \Delta_6\sum_{k=0}^{T-1}\sum_{j=k_0}^{k-1}(1 + \frac{1}{K})^{k-j}w_j\alpha_j^2\left(\sigma_\xi^2 + \sigma_\psi^2\right)$$
$$\le 2\Delta_5 K^2\sum_{k=0}^{T-1}\alpha_k^2 w_k\Xi_k + 2\Delta_6 K\sum_{k=0}^{T-1} w_k\alpha_k^2\left(\sigma_\xi^2 + \sigma_\psi^2\right), \tag{56}$$

where the first inequality is by the choice of $w_k = k + \kappa$ and the fact that $(1 + \frac{1}{K})^{k-j} \le 2$. Substituting the results from (56) into (54) we obtain

$$\sum_{k=0}^{T-1} w_k(\Xi_k + \Phi_k) \le \frac{1}{2}(\kappa - 1)^2\Xi_0^2 + \left(\frac{4\Delta_1}{\lambda_f} + 1\right)\sum_{k=0}^{T-1} w_k\Phi_k + \frac{4\Delta_2}{\lambda_f}\left(\sigma_\xi^2 + \sigma_\psi^2\right)\sum_{k=0}^{T-1} w_k\alpha_k$$
$$\le \frac{1}{2}(\kappa - 1)^2\Xi_0^2 + \left(\frac{4\Delta_1}{\lambda_f} + 1\right)2\Delta_5 K^2\sum_{k=0}^{T-1}\alpha_k^2 w_k\Xi_k$$
$$+ \left(\frac{4\Delta_1}{\lambda_f} + 1\right)2\Delta_6 K\sum_{k=0}^{T-1} w_k\alpha_k^2\left(\sigma_\xi^2 + \sigma_\psi^2\right) + \frac{4\Delta_2}{\lambda_f}\left(\sigma_\xi^2 + \sigma_\psi^2\right)\sum_{k=0}^{T-1} w_k\alpha_k. \tag{57}$$

The above inequality (57) can be simplified as

$$\sum_{k=0}^{T-1} w_k(\Xi_k + \Phi_k) \leq (\kappa - 1)^2 \Xi_0^2$$

$$+ \left(\frac{4\Delta_1}{\lambda_f} + 1\right) 4\Delta_6 K \sum_{k=0}^{T-1} w_k \alpha_k^2 \left(\sigma_\xi^2 + \sigma_\psi^2\right) + \frac{8\Delta_2}{\lambda_f} \left(\sigma_\xi^2 + \sigma_\psi^2\right) \sum_{k=0}^{T-1} w_k \alpha_k, \quad (58)$$

where the inequality is by the choice of step size satisfying $\left(\frac{4\Delta_1}{\lambda_f} + 1\right) 2\Delta_5 K^2 \alpha_k^2 \leq \frac{1}{2}$. Similarly, with $W_T = \sum_{k=0}^{T-1} w_k$, we have that $W_T = \sum_{k=0}^{T-1} w_k = \frac{(2\kappa+T-1)T}{2} \geq \frac{T^2}{2}$. Dividing both sides of (58) by $W_T$ and rearranging the terms, we obtain

$$\frac{1}{W_T} \sum_{k=0}^{T-1} w_k(\Xi_k + \Phi_k) \leq \mathcal{O}\left(\frac{1}{T} + \frac{K^2}{T^2}\right). \quad (59)$$

where we omit the constants involved are only numerical constants that are independent of $K$ and $T$. Since $||y_k - y^*(x_k)||^2 + ||x_k - x^*||^2 \leq \mathcal{O}(\Xi_k + \Phi_k)$, we prove the first part of Theorem 2. Then, Robbins-Siegmund's theorem which together with the choice of $\alpha_k = \Theta(\frac{1}{k+K})$ and $\beta_k = \Theta(\frac{1}{k+K})$ immediately implies that $\lim_{k\to\infty} ||\bar{y}_k - y^*(\bar{x}_k)||^2 = 0 \ a.s.$, $\lim_{k\to\infty} ||\bar{x}_k - x^*||^2 = 0 \ a.s.$, $\lim_{k\to\infty} ||\bar{y}_k - y_{k,i}||^2 = 0 \ a.s.$, and $\lim_{k\to\infty} ||\bar{x}_k - x_{k,i}||^2 = 0 \ a.s.$ □

# D. MISSING PROOFS FOR ALGORITHM 3

We first prove lemma 4, which shall be used to prove Theorem 3.

## D.1. PROOF OF LEMMA 4

*Proof of Lemma 4.* By Assumption 4, for any $k \geq 0$, we have

$$||d_k||^2 = ||\sum_{i=1}^{\tau} \alpha_{k-i} f_{k-i}||^2 = ||\sum_{i=1}^{\tau} \alpha_{k-i} \left(f(x_{k-i}, y_{k-i}) + \xi_{k-i}\right)||^2$$

$$\leq \tau \sum_{i=1}^{\tau} \alpha_{k-i}^2 ||f(x_{k-i}, y_{k-i})||^2 + \sum_{i=1}^{\tau} \alpha_{k-i}^2 \sigma_\xi^2. \quad (60)$$

By Assumption 2 we obtain

$$||f(x_{k-i}, y_{k-i})||^2 \leq 4L_f^2 ||\bar{y}_{k-i} - \bar{y}_{k-i}^*||^2 + 4L^2 ||\bar{x}_{k-i} - x^*||^2 + 4L_f^2 (||d_{k-i}||^2 + ||e_{k-i}||^2). \quad (61)$$

Substituting the results from (61) into (60) gives

$$||d_k||^2 \leq 4L_f^2 \tau \sum_{i=1}^{\tau} \alpha_{k-i}^2 ||\bar{y}_{k-i} - \bar{y}_{k-i}^*||^2 + 4L^2 \tau \sum_{i=1}^{\tau} \alpha_{k-i}^2 ||\bar{x}_{k-i} - x^*||^2$$

$$+ 4L_f^2 \tau \sum_{i=1}^{\tau} \alpha_{k-i}^2 \left(||d_{k-i}||^2 + ||e_{k-i}||^2\right) + \sum_{i=1}^{\tau} \alpha_{k-i}^2 \sigma_\xi^2. \quad (62)$$

Similarly, by Assumption 4, for any $k \geq 0$, we have

$$||e_k||^2 = ||\sum_{i=1}^{\tau} \beta_{k-i} g_{k-i}||^2 = ||\sum_{i=1}^{\tau} \beta_{k-i} \left(g(x_{k-i}, y_{k-i}) + \psi_{k-i}\right)||^2$$

$$\leq \tau \sum_{i=1}^{\tau} \beta_{k-i}^2 ||g(x_{k-i}, y_{k-i})||^2 + \sum_{i=1}^{\tau} \beta_{k-i}^2 \sigma_\psi^2. \quad (63)$$

By Assumption 2, we obtain

$$
\begin{aligned}
||g(x_{k-i}, y_{k-i})||^2 &\leq ||g(x_{k-i}, y_{k-i}) - g(\bar{x}_{k-i}, y_{k-i}) + g(\bar{x}_{k-i}, y_{k-i}) \\
&\quad - g(\bar{x}_{k-i}, \bar{y}_{k-i}) + g(\bar{x}_{k-i}, \bar{y}_{k-i}) - g(\bar{x}_{k-i}, \bar{y}_{k-i}^*)||^2 \\
&\leq 3L_g^2 ||\bar{y}_{k-i} - \bar{y}_{k-i}^*||^2 + 3L_g^2 ||d_{k-i}||^2 + 3L_g^2 ||e_{k-i}||^2.
\end{aligned} \tag{64}
$$

Substituting the results from (64) into (63) yields

$$
\begin{aligned}
||e_k||^2 &\leq 3L_g^2 \tau \sum_{i=1}^{\tau} \beta_{k-i}^2 ||\bar{y}_{k-i} - \bar{y}_{k-i}^*||^2 \\
&\quad + 3L_g^2 \tau \sum_{i=1}^{\tau} \beta_{k-i}^2 \left( ||d_{k-i}||^2 + ||e_{k-i}||^2 \right) + \sum_{i=1}^{\tau} \beta_{k-i}^2 \sigma_\psi^2.
\end{aligned} \tag{65}
$$

Combining the results from (62) and (65), and using the definitions of $\Xi_k$ and $\Phi_k$, we get

$$
\begin{aligned}
\Phi_k &\leq (4L_f^2 + 4L^2 + 3L_g^2 \bar{\beta}^2)\tau \sum_{j=k-\tau}^{k-1} \alpha_j^2 \Phi_j \\
&\quad + (4L_f^2 + 3L_g^2 \bar{\beta}^2)\tau \sum_{j=k-\tau}^{k-1} \alpha_j^2 \Xi_j + (1 + \bar{\beta}^2) \sum_{j=k-\tau}^{k-1} \alpha_j^2 \left( \sigma_\xi^2 + \sigma_\psi^2 \right).
\end{aligned} \tag{66}
$$

Recursively substituting every $\Phi_j$ for $j \geq k - \tau$ we obtain

$$
\begin{aligned}
\Phi_k &\leq (4L_f^2 + 3L_g^2 \bar{\beta}^2)\tau \prod_{t=k-\tau}^{k-1} \left( 1 + (4L_f^2 + 4L^2 + 3L_g^2 \bar{\beta}^2)\tau \alpha_t^2 \right) \sum_{j=k-\tau}^{k-1} \alpha_j^2 \Xi_j \\
&\quad + (1 + \bar{\beta}^2) \prod_{t=k-\tau}^{k-1} \left( 1 + (4L_f^2 + 4L^2 + 3L_g^2 \bar{\beta}^2)\tau \alpha_t^2 \right) \sum_{j=k-\tau}^{k-1} \alpha_j^2 \left( \sigma_\xi^2 + \sigma_\psi^2 \right).
\end{aligned} \tag{67}
$$

Observe that

$$
\prod_{t=k-\tau}^{k-1} \left( 1 + (4L_f^2 + 4L^2 + 3L_g^2 \bar{\beta}^2)\tau \alpha_t^2 \right) \leq \left( 1 + \frac{1}{\tau} \right)^\tau \leq 2, \tag{68}
$$

where the inequality is by the choice of $\alpha_k \leq \frac{1}{\sqrt{(4L_f^2 + 4L^2 + 3L_g^2 \bar{\beta}^2)\tau}}$. Then, (68) can be simplified as

$$
\Phi_k \leq (8L_f^2 + 6L_g^2 \bar{\beta}^2)\tau \sum_{j=k-\tau}^{k-1} \alpha_j^2 \Xi_j + 2(1 + \bar{\beta}^2) \sum_{j=k-\tau}^{k-1} \alpha_j^2 \left( \sigma_\xi^2 + \sigma_\psi^2 \right), \tag{69}
$$

This completes the proof. $\qquad\square$

## D.2. PROOF OF THEOREM 3

*Proof of Theorem 1.* By Lemma 1, we have

$$
\Xi_{k+1} \leq \left( 1 - \frac{\lambda_f}{2} \alpha_k \right) \Xi_k + \Delta_1 \alpha_k \Phi_k + \Delta_2 \alpha_k^2 (\sigma_\xi^2 + \sigma_\psi^2). \tag{70}
$$

Multiplying both sides of (70) by $\frac{4w_k}{\lambda_f \alpha_k}$ and rearranging the terms, we obtain

$$
\begin{aligned}
\sum_{k=0}^{T-1} w_k \Xi_k &\leq 4 \sum_{k=0}^{T-1} \left( \left( 1 - \frac{\lambda_f}{4} \alpha_k \right) \frac{w_k \Xi_k}{\lambda_f \alpha_k} - \frac{w_k \Xi_{k+1}}{\lambda_f \alpha_k} \right) + \frac{4\Delta_1}{\lambda_f} \sum_{k=0}^{T-1} w_k \Phi_k \\
&\quad + \frac{4\Delta_2}{\lambda_f} (\sigma_\xi^2 + \sigma_\psi^2) \sum_{k=0}^{T-1} w_k \alpha_k.
\end{aligned} \tag{71}
$$

Let $\alpha_k = \frac{8}{\lambda_f(k+\kappa)}$ and $w_k = k + \kappa$ where $\kappa \geq 4\tau$ and $W_T = \sum_{k=0}^{T-1} w_k$. Observe that

$$(1 - \frac{\lambda_f\alpha_k}{4})\frac{w_k}{\lambda_f\alpha_k} = \frac{1}{8}(\kappa + k - 2)(\kappa + k) \leq \frac{1}{8}((\kappa + k - 1)^2 - 1) \leq \frac{1}{8}(\kappa + k - 1)^2. \quad (72)$$

Then, the above inequality (71) can be simplified as

$$\sum_{k=0}^{T-1} w_k\Xi_k \leq \frac{1}{2}\sum_{k=0}^{T-1}\left((\kappa + k - 1)^2\Xi_k - (\kappa + k)^2\Xi_{k+1}\right) + \frac{4\Delta_1}{\lambda_f}\sum_{k=0}^{T-1} w_k\Phi_k$$

$$+ \frac{4\Delta_2}{\lambda_f}(\sigma_\xi^2 + \sigma_\psi^2)\sum_{k=0}^{T-1} w_k\alpha_k$$

$$\leq \frac{1}{2}(\kappa - 1)^2\Xi_0^2 + \frac{4\Delta_1}{\lambda_f}\sum_{k=0}^{T-1} w_k\Phi_k + \frac{4\Delta_2}{\lambda_f}(\sigma_\xi^2 + \sigma_\psi^2)\sum_{k=0}^{T-1} w_k\alpha_k. \quad (73)$$

By Lemma 4, we have

$$\Phi_k \leq \Delta_5\tau\sum_{j=k-\tau}^{k-1}\alpha_j^2\Xi_j + \Delta_6\sum_{j=k-\tau}^{k-1}\alpha_j^2\left(\sigma_\xi^2 + \sigma_\psi^2\right), \quad (74)$$

where $\Delta_5 = (8L_f^2 + 6L_g^2\bar{\beta}^2)$ and $\Delta_6 = 2(1 + \bar{\beta}^2)$. Summing (74) over $k = 0$ to $T - 1$ yields

$$\sum_{k=0}^{T-1} w_k\Phi_k \leq 2\Delta_5\tau^2\sum_{k=0}^{T-1}\alpha_k^2 w_k\Xi_k + 2\Delta_6\tau\sum_{k=0}^{T-1} w_k\alpha_k^2\left(\sigma_\xi^2 + \sigma_\psi^2\right), \quad (75)$$

where the inequality is by the choice of $w_k = k + \kappa$ where $\kappa \geq 4\tau$ and the fact that $(1 + \frac{1}{\tau})^{k-j} \leq 2$. Substituting the results from (75) into (73) yields

$$\sum_{k=0}^{T} w_k(\Xi_k + \Phi_k) \leq (\kappa - 1)^2\Xi_0^2 + (1 + \frac{4\Delta_1}{\lambda_f})4\Delta_6\tau\sum_{k=0}^{T-1} w_k\alpha_k^2(\sigma_\xi^2 + \sigma_\psi^2)$$

$$+ \frac{8\Delta_2}{\lambda_f}\sum_{k=0}^{T-1} w_k\alpha_k(\sigma_\xi^2 + \sigma_\psi^2). \quad (76)$$

where the inequality is by the choice of step size satisfying $\left(\frac{4\Delta_1}{\lambda_f} + 1\right)2\Delta_5\tau^2\alpha_k^2 \leq \frac{1}{2}$. Similarly, with $W_T = \sum_{k=0}^{T-1} w_k$, we have that $W_T = \sum_{k=0}^{T-1} w_k = \frac{(2\kappa+T-1)T}{2} \geq \frac{T^2}{2}$. Dividing both sides of (76) by $W_T$ and rearranging the terms, we obtain

$$\frac{1}{W_T}\sum_{k=0}^{T-1} w_k(\Xi_k + \Phi_k) \leq \mathcal{O}\left(\frac{1}{T} + \frac{\tau^2}{T^2}\right). \quad (77)$$

where we omit the constants involved are only numerical constants that are independent of $\tau$ and $T$. Since $||y_k - y^*(x_k)||^2 + ||x_k - x^*||^2 \leq \mathcal{O}(\Xi_k + \Phi_k)$, we prove the first part of Theorem 3. Then, Robbins-Siegmund's theorem which together with the choice of $\alpha_k = \Theta(\frac{1}{k+\tau})$ and $\beta_k = \Theta(\frac{1}{k+\tau})$ immediately implies that $\lim_{k\to\infty}||y_k - y^*(x_k)||^2 = 0 \; a.s.$, and $\lim_{k\to\infty}||x_k - x^*||^2 = 0 \; a.s.$ $\quad\square$