# OpenReview forum: "Error-Feedback Meets Stochastic Approximation with Two Time Scales"
_ICLR.cc/2024/Conference — ICLR 2024 Conference Withdrawn Submission_

### Official Review · Reviewer_3q7e · 2023-10-27

**Soundness:** 3 good
**Presentation:** 3 good
**Contribution:** 2 fair
**Rating:** 5
**Confidence:** 3

**Summary:**

This paper analyzes the convergence of error-feedback two-time-scale SA, and apply their results to three typical application scenarios: error-compensated TTSA with arbitrary compressors, local TTSA with periodic global averaging, and TTSA with delayed updates. In all cases, similar rates $O(1/T+1/T^2)$ are derived, where only the second term is affected by structured perturbations.

**Strengths:**

1. The paper is well-organized and self-contained. The structure is very clear, starting from a generic convergence lemma and delving into three case studies.
2. The originality and motivation is good. The author considers the error-feedback setting in TTSA, which is not studied in literature.

**Weaknesses:**

1. Some definitions and deductions can be further explained to be more reader-friendly. For example,
 - it would be better to explain that $\bar{x_k}, \bar{y_k}$ are TTSA iterates without error feedback after Assumption 4.
 - In Assumption 4, the r.v.s should be $\xi_k, \psi_k$.
 - Adding explanation and the assumptions and lemmas used after deductions in Appendix would be better, e.g. Lemma 6 (7)(8). (How is $\nabla y^*(\bar{x}_k)$ controlled by $\|\xi\|$ in (7)?)
2. Although the paper is self-contained and the motivation of problem is clear, the content is not as much and is entirely restricted in the EF-TTSA convergence analysis. In addition, comparisons with one-time-scale SA or with non-error-feedback may be desired. It would be better if the authors could add some simulation studies to illustrate the difference between EF-TTSA convergence performance and the above mentioned settings, so as to show the significance of the analysis.

**Questions:**

Discussed above.

---

### Official Review · Reviewer_hjVi · 2023-10-28

**Soundness:** 3 good
**Presentation:** 3 good
**Contribution:** 2 fair
**Rating:** 5
**Confidence:** 4

**Summary:**

This paper studies error-feedback-based two-time-scale stochastic approximation. The authors consider three types of structured perturbations including compression, local updates, and delays, and establish non-asymptotic convergence rates. In particular, the leading term $O(1/T)$ in the convergence rate is not affected by the error terms, demonstrating the robustness of two-time-scale stochastic approximation to these structured perturbations.

**Strengths:**

1. The paper is well-organized and easy to follow.
2. To the best of my knowledge, this is the first work to show the robustness of two-time-scale stochastic approximation to structured perturbations.

**Weaknesses:**

1. The title is somewhat exaggerated. In the theoretical analysis, $\beta_k / \alpha_k$ remains a constant. This is only a single-time-scale special case of TTSA. It would be more appropriate to use similar wording to that in [1], e.g., single-time-scale stochastic approximation with two coupled sequences.
2. The assumptions are too strong. For example, this paper only considers the strongly monotone case, while the two previous works [1,2] also study non-strongly monotone cases. The requirement on noise is also restrictive (see Question 1).
3. The proof novelty is somewhat restricted as it heavily relies on the proof techniques in the literature, e.g., those in [1,2].

[1] Han Shen and Tianyi Chen. A single-timescale analysis for stochastic approximation with multiple
coupled sequences. Advances in Neural Information Processing Systems, 35:17415–17429, 2022

[2] Sebastian U Stich and Sai Praneeth Karimireddy. The error-feedback framework: Better rates for
sgd with delayed gradients and compressed updates. The Journal of Machine Learning Research,
21(1):9613–9648, 2020.

Some typos:
* Line 8 in Section 1: 'robustness to robust to'
* The second paragraph in Section 1: TTSA was introduced in (Borkar, 1997).

**Questions:**

1. Assumption 4 requires the noise to have a bounded norm. However, it is generally assumed that the noise has bounded second-order moments. What is the reason for making such a restrictive assumption?
Moreover, Assumption 4 also requires the noise to have zero mean, while in some examples, e.g. stochastic bilevel optimization, it is almost impossible to satisfy such a condition. It would be more appropriate to assume the bias of noise is bounded in some sense, e.g., by the square root of the step size in [1].

2. All the convergence rates in this paper are in terms of the weighted sum of squared norms, while for unbiased SA or TTSA, the last-iterate convergence is achievable. Is this an inevitable issue when there exist structured errors? Or how does the last iterate behave in this case?

[1] Han Shen and Tianyi Chen. A single-timescale analysis for stochastic approximation with multiple
coupled sequences. Advances in Neural Information Processing Systems, 35:17415–17429, 2022

---

### Official Review · Reviewer_FQA6 · 2023-10-31

**Soundness:** 3 good
**Presentation:** 3 good
**Contribution:** 2 fair
**Rating:** 3
**Confidence:** 4

**Summary:**

This paper considers error-feedback based two-timescale stochastic approximatio (EF-TTSA) algorithms; and derive convergence bounds for the EF-TTSA algorithm. It later discusses the applications of the results to explain different variants of EF-TTSA, including error-compensated TTSA with compression, local TTSA with periodic global averaging, and TTSA with delayed updates.

**Strengths:**

+ Convergence results for the so-called unifying EF-TTSA framework.
+ Applications of the results to different instances of EF-TTSA in ML/RL.

**Weaknesses:**

- The paper presents a mixture of different results and tries to explain them all in a so-called unifying EF-TTSA framework. These results can be illusive and make the readers difficult to follow the key idea/contributions of the paper.
- Why shall one study error-feedback in the two-timescale setting is not well-motivated. I understand that stochastic bilevel optimization and stochastic compositional optimization can be cast as special cases of or solved by two-timescale SA algorithms in equations (11-22). Yet, how and where does the error feedback come into the picture? It seems also difficult to intepret the two applications/algrorithms as (23-26). Moreover, in (23-26), it would be clearer if what each of the symbols {x_k,y_k,d_k,e_k} stands or represents in the context of error-feedback SA is explained first. The set of equations in (23-26) looks more like "a four-timescale SA". It is not clear how the EF-TTSA differentiates it from a four-timescale algorithm like this, and is there any advantage treating it as a EF-TTSA?
- Large-scale numerical tests should have been provided to demonstrate the effectiveness of the EF for TTSA as well as motivate the study of the paper.

**Questions:**

1) In Assumption 4, the authors cited "we make the following standard assumption Doan (2022)". Nonetheless, in assumption 4 of Doan (2022), the variables are only assumed zero-mean and with covariances; no bounded assumptions as in (27) have been made. I am not sure whether this can still account for the two instances in Section 2 or not. Furthermore, the paper defines the filtration F_k using only the estimate sequences {x_k} and {y_k} up to time k, which is different than that made in Doan (2022) (which takes also the random variables in the filtration, and assumes the conditional zero mean random variables). Please explain how the random variables are taken care of in the filtration analysis.
2) In Section 4, Assumption 5, the compression operator Q is assumed to satisfy the condition (31) which is a not a mild assumption. It is not fair to claim the results hold for "arbitrary compression".

---

### Meta-Review · Area_Chair_vqsr · 2023-12-14

**Metareview:**

No reviewers proposed acceptance and the authors did not write any rebuttal. The case is clear; there is no need to elaborate.

**Justification For Why Not Higher Score:**

No reviewers proposed acceptance and the authors did not write any rebuttal. The case is clear; there is no need to elaborate.

**Justification For Why Not Lower Score:**

N/A

---

### Decision · Program_Chairs · 2024-01-16

Reject